# Neuro-Symbolic VAEs for Temporal Point Processes: Logic-Guided Controllable Generation

## Abstract

In safety-critical domains such as healthcare, sequential data (e.g., patient trajectories in electronic health records) are often sparse, incomplete, and privacy-sensitive, limiting their utility for downstream modeling. Synthetic sequence generation can mitigate these issues by imputing missing histories and synthesizing new trajectories. However, generation must respect domain constraints to ensure reliability. We propose the Neuro-Symbolic Variational Autoencoder with Temporal Point Processes (NS-VAE-TPP), a framework for logic-aware sequence generation in continuous time. NS-VAE-TPP combines a temporal point process backbone for modeling event times and types with a novel reasoning layer in the latent space. The encoder maps raw streams to high-level predicate variables, while forward-chaining inference enforces logical consistency and imputes missing structure, enabling reliable generation under data scarcity. Symbolic rules are specified as predicate embeddings and enforced as constraints, with flexibility enhanced by querying and refining rule embeddings using large language models (LLMs). Experiments on synthetic data, LogiCity, MIMIC-IV, EPIC-100, and IKEA ASM demonstrate that NS-VAE-TPP achieves more accurate, controllable, and reliable sequence generation under scarce data conditions, highlighting the potential of neuro-symbolic approaches for robust modeling in safety-critical domains.

## 1 Introduction

In many safety-critical domains—such as healthcare, finance, and security—the availability of real-world event sequences (e.g., patient trajectories in electronic health records, activity logs, or financial transactions) is severely limited (Potluru et al., 2023; Wornow et al., 2024). Data are often scarce, privacy-sensitive, and difficult to share; even when accessible, sequences tend to be noisy or incomplete, with missing events that reduce their reliability. This scarcity of high-quality data poses a significant bottleneck for downstream modeling, since modern learning methods typically depend on large, diverse, and trustworthy datasets.

Generative models provide a natural remedy. They can (i) improve data quality by imputing missing events and repairing incomplete histories, and (ii) increase data availability by synthesizing novel, realistic trajectories that expand coverage. Yet in safety-critical settings, synthetic sequences must also respect *domain constraints* to ensure plausibility and reliability. Enforcing constraints only at the output level is insufficient: effective generation requires *reasoning within constraints*, ensuring that latent representations themselves remain consistent even when observations are sparse or incomplete.

To this end, we introduce **Neuro-Symbolic Variational Autoencoder with Temporal Point Processes (NS-VAE-TPP)**, a generative framework that combines the expressiveness of *temporal point processes (TPPs)* with a novel *reasoning layer* in the latent space. TPPs serve as a natural backbone for event sequences, as they capture both event types and times in continuous time, accommodating irregularly spaced observations. Our key contribution is a *reasoning-before-generation* procedure: before sampling, latent representations are refined via forward-chaining inference, which enforces logical consistency and imputes missing structure. By integrating symbolic reasoning directly into the latent space, NS-VAE-TPP produces sequences that are not only constrained at the output level but also internally coherent, reducing spurious correlations and yielding more trustworthy data for downstream tasks.

Recent work has developed TPP-based generative models, showing their ability to simulate realistic sequences (Xiao et al., 2017; Li et al., 2018; Shchur et al., 2019; Mehrasa et al., 2019; Lin et al., 2022). However, these approaches often *degrade under missingness, provide limited control over generation, and lack interpretability*, making it difficult to ensure that application-specific requirements are met.

To address these challenges, our NS-VAE-TPP introduces a *reasoning layer* that incorporates domain knowledge through symbolic rules. For example, $\texttt{Treatment}(x) \leftarrow \texttt{Indication}(x)$ ("a treatment which might be given only if its indication is present"), or $\neg\texttt{Treatment}(x) \leftarrow \texttt{Contraindication}(x)$ ("a treatment must not be given if a contraindication holds").

Each rule is encoded as a *list of predicate embeddings* that act as priors in the latent space. Representing rules in a *continuous space* yields three advantages: (i) reasoning is *differentiable*, (ii) rules integrate naturally into the VAE's latent structure, and (iii) the model performs robust inference under noise or missing data via *soft symbolic matching*.

Specifically, the encoder maps raw event histories into *predicate-level latent variables*, capturing high-level abstractions of the sequence. The reasoning layer then applies *forward-chaining inference* over these variables, enabling the model to impute missing structure, propagate dependencies, and maintain logical consistency among latent states even under sparse observations. Crucially, symbolic rules are not applied as post-hoc constraints; instead, they actively shape the latent space, ensuring that *reasoning precedes generation*. The decoder then samples both event *times* and *types* in continuous time, yielding sequences that are reliable, controllable, and data-efficient.

To support this reasoning process, **NS-VAE-TPP** leverages LLMs to initialize symbolic rules and predicate embeddings. Since LLMs encode rich semantic knowledge of entities and relations, their representations provide ready-made predicate embeddings for the reasoning layer, which can then be fine-tuned during training. This design reduces manual rule engineering, improves adaptability across domains, and offers a practical path toward *scalable neuro-symbolic sequence generation*.

## 1.1 LITERATURE REVIEW

Generative models have achieved notable success in domains such as healthcare (Choi et al., 2017; Biswal et al., 2021; Zhang et al., 2021), finance (Assefa et al., 2020; Dogariu et al., 2022), and vision (Tulyakov et al., 2018; Yang et al., 2023; Ho et al., 2022), producing synthetic datasets that closely mimic real data. The advent of LLMs has further expanded generative capabilities, particularly in text (Brown et al., 2020; Thoppilan et al., 2022; Chowdhery et al., 2023) and even EHR synthesis (Theodorou et al., 2023), showcasing unprecedented flexibility.

Yet most approaches assume fixed sampling grids and ignore *irregular inter-event times*—such as bursts or long gaps—that are central to real-world event streams. TPPs (Daley & Vere-Jones, 2007; Reynaud-Bouret & Schbath, 2010) directly address this challenge by modeling both event *times* and *types* in continuous time, and have become standard tools for fine-grained sequence modeling (Enguehard et al., 2020). Building on this foundation, deep generative TPPs include VAE formulations that parameterize conditional laws or intensities (Mehrasa et al., 2019; Lin et al., 2022), Wasserstein-based objectives that minimize distributional distance (Xiao et al., 2017), RL-based formulations with MMD-style rewards (Li et al., 2018), and intensity-free samplers that bypass explicit parameterization (Shchur et al., 2019). While effective, these models remain largely *black-box*, offering limited interpretability and control over generated sequences.

To mitigate these limitations, recent work has explored combining LLMs with TPPs to improve reasoning and interpretability in sequential modeling (Liu & Quan, 2024; Shi et al., 2023; Song et al., 2024). However, LLM knowledge is typically used *post hoc* for explanation or prediction rather than embedded directly as a *generative prior*. As a result, symbolic constraints are not consistently propagated throughout the continuous-time sampling path, limiting the ability to enforce *domain-specific rules* during generation.

This limitation is critical. In high-stakes domains, producing statistically realistic samples alone is insufficient. Synthetic sequences must also satisfy *symbolic constraints*—such as eligibility, exclusions, ordering, and timing—that encode dependencies beyond surface correlations. Enforcing such rules not only improves trustworthiness but also ensures that generated data are actionable and domain-valid.

Prior work has sought to combine rules with TPPs, for instance by embedding symbolic constraints into intensity functions (Li et al., 2020b; 2021), applying EM-style inference to uncover latent rules (Kuang et al., 2024), or using neuro-symbolic methods for temporal rule extraction (Yang et al., 2024) and structured synthesis in non-sequential domains (Li et al., 2024b). While promising, these

approaches primarily focus on *rule mining*. They rarely use symbolic knowledge as a *generative prior*, and typically lack the *multi-hop reasoning* needed to infer latent high-level concepts or impute missing structures from partial histories. As a result, *constraint-aware, concept-conditional generation* remains underexplored.

**Summary of Contributions.**   We propose **NS-VAE-TPP**, a neuro-symbolic generative framework for constraint-aware sequence modeling in continuous time. Our contributions are: (i) a *reasoning-before-generation* architecture that embeds *symbolic rules as generative priors* in the latent space, enabling logical consistency, structure imputation, and constraint-aware generation beyond post-hoc rule enforcement; (ii) the use of *LLM-initialized predicate embeddings and rules*, which reduces manual effort in rule engineering and improves adaptability across domains; and (iii) extensive experiments on synthetic, semi-synthetic, and real-world datasets demonstrating improved reliability, controllability, and data efficiency under scarce and incomplete observations.

## 2 BACKGROUND KNOWLEDGE

### 2.1 PREDICATES, LOGIC RULES, AND FORWARD CHAINING

We define a **predicate** as a Boolean function $P : \mathcal{X} \to \{0, 1\}$, where $\mathcal{X}$ denotes the space of observed event sequences. Predicates abstract raw events into high-level symbolic concepts that can be reasoned over. Let $\mathcal{P}$ denote the set of all predicates.

Domain knowledge is encoded as a *rule set* $\mathcal{F}$, where each $f \in \mathcal{F}$ is a definite **Horn clause**:

$$f : \quad P_0(x) \leftarrow P_1(x) \wedge P_2(x) \wedge \cdots \wedge P_h(x), \quad x \in \mathcal{X}, \tag{1}$$

with $P_0 \in \mathcal{P}$ the *head* and $P_1, \ldots, P_h \in \mathcal{P}$ the *body*. Intuitively, $P_0(x)$ is inferred whenever all body predicates hold for the same $x$. Each rule specifies how new facts (unobserved predicates) can be derived from observed or previously inferred ones.

Let $\Gamma_0 \subseteq \mathcal{P}$ be the set of *grounded predicates* observed from data. **Forward chaining** iteratively adds new facts:

$$\Gamma_{h+1} = \Gamma_h \cup \Big\{ P_0(x) \mid (P_0(x) \leftarrow P_1(x) \wedge \cdots \wedge P_h(x)) \in \mathcal{F}, \ P_1(x), \ldots, P_h(x) \in \Gamma_h \Big\}, \tag{2}$$

until no new facts are added, producing the closure $\Gamma^\star$. This implements *multi-hop reasoning*, deriving higher-level predicates from grounded observations.

### 2.2 TEMPORAL POINT PROCESSES (TPPs)

A TPP models sequences of events $(t_i, m_i)$, where $t_i \in \mathbb{R}^+$ is the event time and $m_i \in \mathcal{M}$ is the event type. Let $x_t = \{(t_j, m_j) \mid t_j < t\}$ denote the sequence history up to time $t$.

The dynamics of the TPP are described by the conditional intensity function $\lambda(t, m \mid x_t) = \lim_{\Delta t \to 0} \frac{\mathbb{P}(\text{event of type } m \text{ in } [t, t+\Delta t) \mid x_t)}{\Delta t}$, with total intensity $\lambda(t \mid x_t) = \sum_{m \in \mathcal{M}} \lambda(t, m \mid x_t)$.

From a generative perspective, a TPP generates events sequentially. The next event time is drawn from the survival function, and then the event type is sampled conditional on that time:

$$\mathbb{P}(T > \tau \mid x_{t_i}) = \exp\Big(-\int_{t_i}^{\tau} \lambda(s \mid x_s)\, ds\Big), \quad \mathbb{P}(m = k \mid t, x_t) = \frac{\lambda(t, k \mid x_t)}{\lambda(t \mid x_t)}. \tag{3}$$

When the integrated intensity does not have a closed-form inverse, Ogata's thinning algorithm (Ogata, 1981; Rasmussen, 2018) provides an efficient way to sample events.

This framework naturally handles *irregularly spaced events and continuous-time dynamics*, making TPPs a flexible foundation for modeling complex event sequences in NS-VAE-TPP.

## 3 NEURO-SYMBOLIC VARIATIONAL AUTOENCODER WITH TEMPORAL POINT PROCESSES (NS-VAE-TPP)

Given observed training event sequences, where each sequence is $x = \{(t_i, m_i)\}_{i=1}^N$, consisting of $N$ events. Each event $(t_i, m_i)$ has a *time* $t_i \in \mathbb{R}^+$ and an *event type* $m_i \in \mathcal{M}$. Our NS-VAE-TPP model aims to generate a synthetic sequence $\hat{x} = \{(\hat{t}_i, \hat{m}_i)\}_{i=1}^{\hat{N}}$ conditioned on initial evidence $x_0$, which may be empty (fully generative) or include partial observations (conditional generation). The objective is to generate $\hat{x}$ that captures the temporal dynamics and event structure of the ground-truth sequence $x$, while adhering to high-level symbolic constraints. This formulation naturally handles irregular sequences, partial observations, and missing events, reflecting realistic challenges in safety-critical domains.

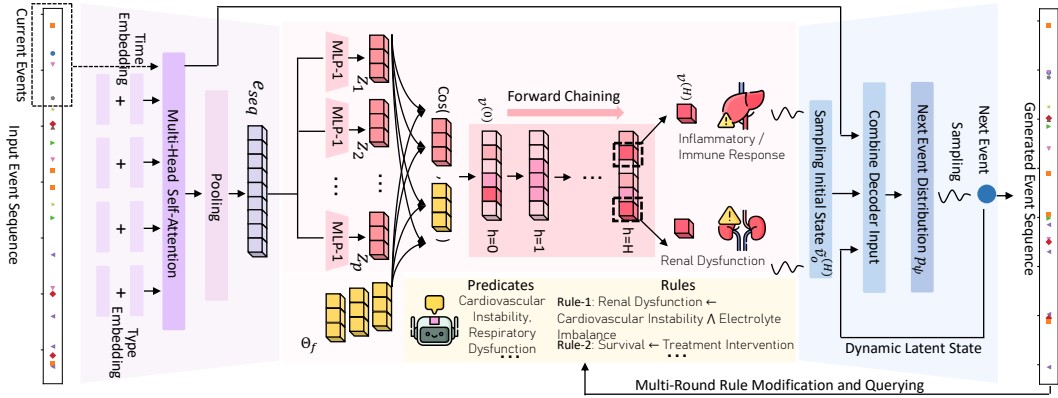

Figure 1: Model framework. The background color indicates: "▮": History Encoder, "▮": Symbolic Prior Bank,"▮": Neuro-Symbolic Reasoning Layer "▮": Event Decoder.

The NS-VAE-TPP model is composed of four interconnected modules:

- **Symbolic Prior Bank (SPB):** provides the global predicate set and initial rule set, derived from an LLM.
- **History Encoder:** maps raw event sequences into temporal embeddings $\boldsymbol{H}_t$, summarizing the sequence up to time $t$.
- **Neuro-Symbolic Reasoning Layer:** integrates the SPB and $\boldsymbol{H}_t$ to compute multi-hop predicate states $\boldsymbol{v}_t^{(H)}$, capturing high-level reasoning over observed events.
- **Event Decoder:** generates the next events conditioned on both $\boldsymbol{H}_t$ and reasoning-augmented latent states.

Each module is detailed in the subsections below.

### 3.1 SYMBOLIC PRIOR BANK (SPB): GLOBAL PREDICATE SET & INITIAL RULES

To inject domain knowledge and symbolic constraints, we query an LLM with a few example sequences and contextual instructions (see Appendix B for prompts). The LLM outputs two key components:

- **Global Predicate Set.**
$$\mathcal{P} = \{P_1, \ldots, P_K\}, \quad |\mathcal{P}| = K,$$
where each predicate $P \in \mathcal{P}$ is assigned a semantic embedding $\theta_P \in \mathbb{R}^d$, extracted from the LLM's internal representations (Nie et al., 2024). The collection of embeddings is
$$\boldsymbol{Z} = \{\theta_P\}_{P \in \mathcal{P}}.$$
These act as *fixed semantic anchors* capturing the meaning of predicates.

- **Initial Rule Set.** Symbolic rules are expressed as Horn clauses:
$$f: \quad P_0(x) \leftarrow P_1(x) \wedge \cdots \wedge P_c(x),$$
where $P_0$ is the head and $P_1, \ldots, P_c$ are body predicates. Each rule $f$ is mapped to a trainable embedding sequence
$$\Theta_f = [\theta_{P_0}, \theta_{P_1}, \ldots, \theta_{P_c}] \in \mathbb{R}^{d \times (c+1)},$$
and the full collection forms the *Symbolic Prior Bank (SPB)*:
$$\Theta_{\mathcal{F}} = \{\Theta_f\}_{f \in \mathcal{F}}.$$

The predicate embeddings $\{\theta_P\}$ remain *fixed* during training, while the rule embeddings $\Theta_{\mathcal{F}}$ are *trainable*. This enables the NS-VAE-TPP to refine how predicates interact in rules, while preserving LLM-defined semantic meanings. Thus, the LLM serves as a *knowledge initializer* rather than a post-hoc explainer, providing constraint-aware, high-level reasoning to the model. In short,
$$\{\text{Global Predicate Set}, \text{ Initial Rule Set}\} = \textbf{LLM}(\text{prompt}).$$

### 3.2 HISTORY ENCODER

The history encoder maps raw event sequences into temporal embeddings that capture both event type and temporal information. Formally, consider a sequence of $N$ events:
$$x = \{(t_i, m_i)\}_{i=1}^N,$$
where each event consists of a timestamp $t_i \in \mathbb{R}^+$ and an event type $m_i \in \mathcal{M}$.

**Sequence Embedding.** Each event $(t_i, m_i)$ is mapped into a continuous vector by combining two components:

$$\boldsymbol{z}_i = \boldsymbol{e}_{m_i}^{\text{type}} + \phi(t_i),$$

where $\boldsymbol{e}_{m_i}^{\text{type}} \in \mathbb{R}^{d_e}$ is a learnable type embedding, and $\phi(t_i) \in \mathbb{R}^{d_e}$ is a time embedding (e.g., sinusoidal or neural projection of $t_i$). The sequence of embeddings $\{\boldsymbol{z}_i\}_{i=1}^N$ is processed by a Transformer encoder (Vaswani et al., 2017; Zuo et al., 2020):

$$\boldsymbol{H} = \text{Encoder}(x) = \text{Transformer}([\boldsymbol{z}_1, \ldots, \boldsymbol{z}_N]) \in \mathbb{R}^{N \times d_e}, \quad (4)$$

yielding contextualized representations for all events. Then, a global history embedding is obtained by pooling:

$$e^{\text{seq}} = \text{Pool}(\boldsymbol{H}) \in \mathbb{R}^{d_e}.$$

Thus, the history encoder output $\boldsymbol{H}$ provides time- and type-aware contextual embeddings, serving as the input to the neuro-symbolic reasoning layer.

### 3.3 NEURO-SYMBOLIC REASONING LAYER

The neuro-symbolic reasoning layer first grounds the predicates from observation, then refines the initial grounded predicates via forward chaining, which is realized by differentiable operators.

**Initial Latent State from History.** Given the sequence embedding $e^{\text{seq}}$, each predicate $P \in \mathcal{P}$ is projected via a predicate-specific MLP:

$$z_P = g_P(e^{\text{seq}}) \in \mathbb{R}^d,$$

and a *soft match* with its fixed semantic embedding $\theta_P$ produces the initial latent vector:

$$\boldsymbol{v}^{(0)} = [v_1, \ldots, v_K] \in [0, 1]^K, \quad v_P = \frac{\cos(z_P, \theta_P) + 1}{2}. \quad (5)$$

$\boldsymbol{v}^{(0)} \in [0, 1]^K$ is the **key hidden state** in our model, summarizing how strongly the observed sequence supports each predicate before reasoning. This vector serves as the initial state for the neuro-symbolic reasoning layer.

**Forward Chaining as a Differentiable Operator.** The reasoning process is implemented as a recursive operator that updates $\boldsymbol{v}^{(h)}$ at hop $h$ using the rule set $\Theta_{\mathcal{F}}$. One hop corresponds to applying all rules once, propagating evidence from bodies to heads. Repeating this process $H$ times produces $H$-hop reasoning.

*Step 1: Rule Grounding.* Each rule embedding $\Theta_f = [\theta_{P_0}, \theta_{P_1}, \ldots, \theta_{P_c}]$ specifies an abstract symbolic pattern. To apply it, we align each slot $\theta_i$ with the closest predicate embedding in the global set $\boldsymbol{Z} = \{\theta_P\}$:

$$z_i^* = \arg \max_{z \in \boldsymbol{Z}} \cos(z, \theta_i), \quad i = 0, \ldots, c. \quad (6)$$

This grounds the rule to concrete predicates. Each matched body predicate $z_i^*$ inherits its current value $v_{z_i^*}^{(h)}$ from the latent state.

*Step 2: Body Aggregation.* Evidence from the body predicates is aggregated into a rule score using a differentiable approximation of logical AND:

$$u_f = \text{softmin}^1\Big(\{\cos(z_i^*, \theta_i),\ v_{z_i^*}^{(h)}\}_{i=1}^c\Big), \quad (7)$$

where $\tau$ in the softmin controls smoothness. The head is activated only when all body predicates are sufficiently supported. This formulation explicitly indicates that each body predicate contributes evidence from two aspects: semantic alignment (cosine similarity) and current confidence level (activation value).

*Step 3: Head Update.* If a predicate $P$ is the head of multiple rules, contributions are aggregated:

$$v_P^{(h+1)} = \text{Agg}\Big(\{u_f : f \in \mathcal{F},\ \text{head}(f) = P\}\Big), \quad (8)$$

with Agg chosen as softmax[2], which is an approximation of logical OR while remaining fully differentiable.

---

[1]Softmin is a smooth approximation of the min function. Here we are using $\text{softmin}(x_1, \ldots, x_c) = -\tau \log \sum_{i=1}^c \exp(-x_i/\tau)$, where $\tau$ is the tuning temperature parameter.

[2]Softmax is a smooth approximation of the max function. Here we are using $\text{softmax}(u_1, \ldots, u_m) = \frac{\sum_{j=1}^m u_j \exp(u_j/\tau)}{\sum_{j=1}^m \exp(u_j/\tau)}$.

*Recursive Form.* Together, one hop of forward chaining is

$$\boldsymbol{v}^{(h+1)} = \text{ForwardChain}(\boldsymbol{v}^{(h)}, \Theta_{\mathcal{F}}) \tag{9}$$

$$= \text{HeadUpdate}\big(\text{BodyAgg}(\text{RuleGround}(\boldsymbol{v}^{(h)}, \Theta_{\mathcal{F}}))\big). \tag{10}$$

**Connection to Classical Forward Chaining.** Recall from the background that forward chaining (Eq. 2) begins from grounded predicates $\Gamma_0 \subseteq \mathcal{P}$ and iteratively adds new facts until convergence, producing the closure $\Gamma^{\star}$. Each iteration corresponds to one reasoning step, deriving new head predicates from already-activated body predicates.

In our setting, forward chaining is realized on a continuous vector representation. Instead of a discrete set $\Gamma_h$, we maintain a predicate-value vector $\boldsymbol{v}^{(h)} \in [0, 1]^K$, where each entry gives the current belief strength of a predicate at iteration (or *hop*) $h$.

Classically, forward chaining runs until convergence, producing $\Gamma^{\star}$. Ideally, we want the differentiable operator would also reach a fixed point. However, increasing the number of reasoning hops enables deeper reasoning, which can improve inference accuracy but often leads to vanishing or exploding gradients. Conversely, using fewer hops results in shallower reasoning that may limit expressive power, yet it ensures more stable and efficient gradient propagation.. In practice, to stabilize optimization and mitigate gradient vanishing issues, we adopt a fixed number of hops $H$. The final state

$$\boldsymbol{v}^{(H)} = \text{ForwardChain}^H(\boldsymbol{v}^{(0)}, \Theta_{\mathcal{F}})$$

serves as an $H$-step approximation to multi-hop reasoning in continuous space. This forward chaining can help *enforce logical consistency* and partially *impute missing knowledge*.

### 3.4 EVENT DECODER

**Discrete Latent Variable.** The NS-VAE-TPP employs a *discrete latent state* $\boldsymbol{v} \in \{0, 1\}^K$, representing the activation of $K$ high-level predicates. The posterior is approximated by a factorized Bernoulli distribution:

$$q(\boldsymbol{v} \mid x) = \prod_{k=1}^{K} \text{Bernoulli}([v^H]_k), \qquad \boldsymbol{v}^H = \text{ForwardChain}(\text{Encoder}(x), \Theta_{\mathcal{F}}), \tag{11}$$

where $[v^H]_k$ denotes the posterior probability of the $k$-th predicate (being 0 or 1) given $x$.

The prior over $\boldsymbol{v}$ is defined as an independent Bernoulli distribution:

$$p(\boldsymbol{v}) = \prod_{k=1}^{K} \text{Bernoulli}(\pi_k), \tag{12}$$

where $\pi_k$ may be uniform or learned, providing a global initialization for predicate activations. The utilization of the Bernoulli distribution enables the KL divergence in the ELBO (later illustrated in Eq. (18)) between the posterior and prior to have a simple closed-form solution. Therefore, it is a deliberate and mathematically convenient choice that enables efficient gradient-based learning of the symbolic rules while naturally matching the binary nature of the predicates, all within a tractable VAE optimization objective.

**Decoder State.** Our model distinguishes two latent states:

- **Global latent state** $\tilde{\boldsymbol{v}}_0^H \in [0, 1]^K$, sampled once at $t = 0$:
$$\tilde{\boldsymbol{v}}_0^H \sim q(\boldsymbol{v}_0^H \mid x_0), \tag{13}$$
where $x_0$ is empty for unconditional generation or contains evidence for conditional generation. This state encodes *global semantic constraints* from the SPB.

- **Dynamic latent state** $\boldsymbol{v}_t^H \in [0, 1]^K$, updated at each time step from the generated history:
$$\boldsymbol{v}_t^H = \text{ForwardChain}(\text{Encoder}(x_{<t}), \Theta_{\mathcal{F}}). \tag{14}$$
This state captures *emerging facts* inferred during sequence generation.

At each step $t$, the decoder input is:

$$s_t = \big[\text{Pool}(\boldsymbol{H}_t); \ \tilde{\boldsymbol{v}}_0^H; \ \boldsymbol{v}_t^H\big], \tag{15}$$

combining temporal history, global constraints, and dynamic reasoning.

**Next-Event Distribution.** Following the TPP formulation (Eq. (3)), we parameterize the conditional distribution of the next event as

$$p_\psi(\Delta t, m \mid s_t) = p_\psi(\Delta t \mid s_t) \ p_\psi(m \mid s_t). \tag{16}$$

Here, $p_\psi(\Delta t \mid s_t)$ is an exponential-family distribution capturing inter-event times, and $p_\psi(m \mid s_t)$ is a softmax over event types $\mathcal{M}$. This design retains the probabilistic semantics of intensity-driven TPPs while avoiding intractable integrations.

**Sampling.** At inference time, the next event is drawn as

$$\hat{\Delta}t_{i+1} \sim p_\psi(\Delta t \mid s_t), \qquad \hat{m}_{i+1} \sim p_\psi(m \mid s_t). \tag{17}$$

### 3.5 GENERATIVE MODEL AND ELBO

The generative process is summarized in Alg. 1, Appendix. A. The training objective is the ELBO:

$$\mathcal{L}_{\text{ELBO}} = \mathbb{E}_{\tilde{\boldsymbol{v}}_0^H \sim q(\boldsymbol{v}_0^H \mid x_0)} \left[ \sum_{i=1}^N \log p_\psi(\Delta t_i, m_i \mid s_i) - \text{KL}\big(q(\boldsymbol{v}_0^H \mid x_0) \,\|\, p(\boldsymbol{v}_0)\big) \right]. \tag{18}$$

The likelihood term is computed autoregressively with both latent states, while the KL term (between Bernoulli distributions) regularizes the initial latent vector. For training, discrete latent samples are relaxed via the Gumbel–Sigmoid reparameterization (Jang et al., 2016) to enable end-to-end gradient-based optimization. This enables joint training of the encoder, decoder, and symbolic priors under the VAE framework.

## 4 EXPERIMENTS

### 4.1 EXPERIMENTAL SETUP

**Datasets** We evaluate NS-VAE-TPP on six event sequence datasets: two synthetic, one semi-synthetic, and three real-world. For synthetic and semi-synthetic datasets, ground-truth logic rules are available and directly provided, enabling neuro-symbolic reasoning without rule inference. For real-world datasets, rules are unknown and are extracted by iteratively querying LLMs. **Synthetic: Syn@5** and **Syn@10** contain sequences of 5 and 10 predicates, respectively, sampled from Temporal Logic Point Processes (TLPP) (Li et al., 2020b). **Semi-synthetic: LogiCity** (Li et al., 2024a) generates multi-agent urban events using customizable first-order logic (FOL) rules. **Real-world: MIMIC-IV** (Johnson et al., 2020) contains ICU patient records, from which sepsis-related event sequences are extracted (Saria, 2018). **EPIC-Kitchen-100** (Damen et al., 2020) provides annotated kitchen action sequences (verbs) to build temporal event histories. **IKEA ASM** (Ben-Shabat et al., 2021) includes human actions during furniture assembly; we focus on TV-bench sequences.

All datasets are split 80%/10%/10% for train/val/test. Additional details are in Appendix C.1.

**Baselines** We compare NS-VAE-TPP against state-of-the-art baselines from two categories. (i) *Neural TPP models*. We include four strong neural TPP models—**AVAE** (Mehrasa et al., 2019), **GNTPP** (Lin et al., 2022), **A & T (Add-and-Thin)** (Lüdke et al., 2023), and **UFM-TPP** (Shou, 2025)—all of which are designed to model event intensities and can generate event sequences by sampling from their learned temporal distributions. These methods represent diverse paradigms including variational inference, graph-based modeling, thinning-based sampling, and unified flow-matching for TPPs. (ii) *LLM-based models*: We also design LLM-based baselines to explore their event sequence generation capability. **SP (Simple-Prompt)**: Inspired by Si et al. (2022), we treat prompts as samplers, directly asking the LLM to output full TPP sequences in the form "$(t_1, \texttt{event}_1), (t_2, \texttt{event}_2), \dots$". **QA (Question-Answering)**: Following the recursive generation strategy of EDGAR (Castricato et al., 2021), the LLM is prompted in a QA manner to predict the next event and its timestamp step by step. **LAMP** (Shi et al., 2023): We adapt LAMP by replacing its original event proposer—which selected individual future events—with a full-sequence candidate generator that produces a complete segment in a single pass. Further implementation details of all baselines are provided in Appendix C.2.

**Evaluation Metrics** We evaluate the generation quality under two regimes:
(i) *In-Distribution Generation*—when the training data sufficiently cover the target distribution or the true data distribution is known (e.g., synthetic/semi-synthetic with oracle rules). We compute **KL Divergence** and **QQ-RMSE** (Xiao et al., 2017) against the true event-time distributions, and **MMD (Maximum Mean Discrepancy)** (Shchur et al., 2020) between generated and real sequences. For *real-world* datasets without oracle rules, we report **DS (Discriminator Score)** (Desai et al., 2021)—defined as ($\texttt{accuracy} - 0.5$) of a 2-layer LSTM trained to distinguish *Real* vs. *Not Real* (values near 0 indicate high realism)—and **GPTScore** (Fu et al., 2023), the likelihood assigned by a pre-trained language model to event-sequence text.

(ii) *Rule-Conditioned (Out-of-Distribution) Generation*—when the training data are insufficient to cover the target distribution or the true distribution is hard to characterize. We use LLM judges to score: **R-Score (Rule Adherence)**: degree to which a sequence satisfies the provided logic rules. **C-Score (Contextual Plausibility)**: judged plausibility of the sequence given domain context under those rules. Both scores lie in $(0, 1)$ and quantify validity beyond what limited data alone can support.

## 4.2 Results and Analysis

**Analysis 1: Generation Performance and Comparison with Baselines** The primary findings are summarized in Tab. 1. The synthetic and semi-synthetic evaluations are designed to measure how effectively our model can leverage given logic rules, not to claim a fair SOTA comparison against baselines that cannot use symbolic inputs. Across these datasets, our method consistently outperforms all baselines on all three generation metrics. This demonstrates that, in a controlled setting, our framework can fully exploit ground-truth rules and allows us to directly quantify the benefit of rule-

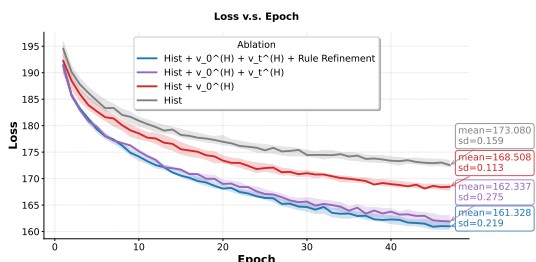

Figure 2: Training loss for different ablation cases.

guided generation. On the real-world MIMIC-IV dataset, our method outperforms the strongest baseline (A & T) by relative margins of $18\%$ (DS), $14\%$ (GPTScore), and $23\%$ (MMD), all statistically significant. It also achieves superior performance on the IKEA dataset. On EPIC-100, our model closely matches A & T, with only a small gap.

| | Synthetic Datasets | | | | | | | | | Semi-Synthetic Datasets | | |
| --- | --- | --- | --- | --- | --- | --- | --- | --- | --- | --- | --- | --- |
| | Syn@5 | | | Syn@10 | | | | | | LogiCity | | |
| **Methods** | KL ↓ | QQ-RMSE ↓ | MMD ↓ | KL ↓ | QQ-RMSE ↓ | MMD ↓ | | | | KL ↓ | QQ-RMSE ↓ | MMD ↓ |
| AVAE | $0.67_{\pm0.04}$ | $0.38_{\pm0.03}$ | $0.62_{\pm0.05}$ | $0.79_{\pm0.05}$ | $0.67_{\pm0.04}$ | $0.81_{\pm0.06}$ | | | | $0.86_{\pm0.08}$ | $0.83_{\pm0.09}$ | $1.05_{\pm0.05}$ |
| GNTPP | $0.62_{\pm0.06}$ | $0.37_{\pm0.03}$ | $0.54_{\pm0.07}$ | $0.71_{\pm0.07}$ | $0.62_{\pm0.03}$ | $0.76_{\pm0.02}$ | | | | $0.80_{\pm0.05}$ | $0.72_{\pm0.04}$ | $0.94_{\pm0.04}$ |
| A & T | $0.46_{\pm0.02}$ | $0.27_{\pm0.01}$ | $0.37_{\pm0.04}$ | $0.67_{\pm0.04}$ | $0.59_{\pm0.02}$ | $0.72_{\pm0.04}$ | | | | $0.70_{\pm0.04}$ | $0.65_{\pm0.05}$ | $0.85_{\pm0.07}$ |
| UFM-TPP | $0.54_{\pm0.04}$ | $0.31_{\pm0.04}$ | $0.48_{\pm0.03}$ | $0.72_{\pm0.05}$ | $0.64_{\pm0.06}$ | $0.70_{\pm0.04}$ | | | | $0.75_{\pm0.07}$ | $0.67_{\pm0.05}$ | $0.88_{\pm0.07}$ |
| SP | $1.12_{\pm0.07}$ | $0.62_{\pm0.06}$ | $1.24_{\pm0.11}$ | $1.14_{\pm0.08}$ | $0.78_{\pm0.07}$ | $1.06_{\pm0.10}$ | | | | $1.05_{\pm0.09}$ | $0.79_{\pm0.06}$ | $1.23_{\pm0.10}$ |
| QA | $0.78_{\pm0.10}$ | $0.40_{\pm0.05}$ | $0.71_{\pm0.12}$ | $0.85_{\pm0.08}$ | $0.72_{\pm0.06}$ | $0.87_{\pm0.07}$ | | | | $0.96_{\pm0.04}$ | $0.85_{\pm0.08}$ | $1.16_{\pm0.11}$ |
| LAMP | $0.51_{\pm0.03}$ | $0.29_{\pm0.01}$ | $0.43_{\pm0.02}$ | $0.68_{\pm0.02}$ | $0.56_{\pm0.04}$ | $0.64_{\pm0.05}$ | | | | $0.74_{\pm0.06}$ | $0.70_{\pm0.04}$ | $0.79_{\pm0.05}$ |
| **Ours\*** | $\mathbf{0.39_{\pm0.03}}$ | $\mathbf{0.22_{\pm0.05}}$ | $\mathbf{0.30_{\pm0.06}}$ | $\mathbf{0.62_{\pm0.07}}$ | $\mathbf{0.53_{\pm0.09}}$ | $\mathbf{0.55_{\pm0.04}}$ | | | | $\mathbf{0.64_{\pm0.02}}$ | $\mathbf{0.63_{\pm0.07}}$ | $\mathbf{0.73_{\pm0.05}}$ |

| | Real-World Datasets | | | | | | | | |
| --- | --- | --- | --- | --- | --- | --- | --- | --- | --- |
| | MIMIC-IV | | | EPIC-100 | | | IKEA ASM | | |
| **Methods** | DS ↓ | GPTScore ↑ | MMD ↓ | DS ↓ | GPTScore ↑ | MMD ↓ | DS ↓ | GPTScore ↑ | MMD ↓ |
| AVAE | $0.46_{\pm0.02}$ | $0.42_{\pm0.03}$ | $0.72_{\pm0.03}$ | $0.45_{\pm0.06}$ | $0.51_{\pm0.01}$ | $1.41_{\pm0.09}$ | $0.48_{\pm0.03}$ | $0.54_{\pm0.02}$ | $0.75_{\pm0.02}$ |
| GNTPP | $0.44_{\pm0.03}$ | $0.53_{\pm0.05}$ | $0.65_{\pm0.04}$ | $0.43_{\pm0.05}$ | $0.51_{\pm0.02}$ | $1.29_{\pm0.08}$ | $0.45_{\pm0.03}$ | $0.58_{\pm0.02}$ | $0.69_{\pm0.04}$ |
| A & T | $0.38_{\pm0.02}$ | $0.64_{\pm0.04}$ | $0.56_{\pm0.03}$ | $\mathbf{0.36_{\pm0.04}}$ | $\mathbf{0.62_{\pm0.01}}$ | $1.03_{\pm0.07}$ | $0.37_{\pm0.02}$ | $0.60_{\pm0.00}$ | $0.63_{\pm0.05}$ |
| UFM-TPP | $0.39_{\pm0.02}$ | $0.55_{\pm0.06}$ | $0.61_{\pm0.03}$ | $0.40_{\pm0.04}$ | $0.53_{\pm0.04}$ | $1.16_{\pm0.08}$ | $0.46_{\pm0.01}$ | $0.52_{\pm0.04}$ | $0.71_{\pm0.03}$ |
| SP | $0.45_{\pm0.10}$ | $0.38_{\pm0.07}$ | $0.94_{\pm0.07}$ | $0.47_{\pm0.05}$ | $0.41_{\pm0.02}$ | $1.63_{\pm0.10}$ | $0.48_{\pm0.07}$ | $0.51_{\pm0.04}$ | $1.38_{\pm0.13}$ |
| QA | $0.42_{\pm0.09}$ | $0.43_{\pm0.05}$ | $0.71_{\pm0.06}$ | $0.47_{\pm0.03}$ | $0.46_{\pm0.02}$ | $1.47_{\pm0.14}$ | $0.47_{\pm0.03}$ | $0.49_{\pm0.05}$ | $0.79_{\pm0.07}$ |
| LAMP | $0.41_{\pm0.02}$ | $0.60_{\pm0.04}$ | $0.58_{\pm0.04}$ | $0.45_{\pm0.02}$ | $0.54_{\pm0.03}$ | $1.06_{\pm0.09}$ | $0.39_{\pm0.02}$ | $0.62_{\pm0.03}$ | $0.64_{\pm0.03}$ |
| **Ours\*** | $\mathbf{0.31_{\pm0.01}}$ | $\mathbf{0.73_{\pm0.06}}$ | $\mathbf{0.43_{\pm0.04}}$ | $0.39_{\pm0.03}$ | $0.58_{\pm0.02}$ | $\mathbf{0.94_{\pm0.12}}$ | $\mathbf{0.32_{\pm0.02}}$ | $\mathbf{0.65_{\pm0.06}}$ | $\mathbf{0.57_{\pm0.04}}$ |

Table 1: Comparison of different methods. Performance metrics are averaged across three different runs, which reported as (Mean $+/-$ SD). The best performance is in bold and also colored in purple.

**Analysis 2: Evidence of Parameter Efficiency, Data Efficiency, and Robustness** The synthetic setting enables a clean assessment of our model's advantages. By incorporating logic rules, our method achieves better complexity handling and reduced data dependence than all baselines. As shown in Fig. 2, neural TPP baselines (e.g., AVAE, GNTPP) require substantially larger models yet still fall short of our performance, highlighting the parameter efficiency gained from structured symbolic priors. Our few-shot results further demonstrate strong data efficiency: on MIMIC-IV (details in Analysis 4), using only $20\%$ of the training data, our model achieves an MMD of 0.54—still outperforming the best baseline A & T trained on the full dataset (MMD: 0.56).

The framework remains robust even with incomplete or missing rule priors. On Syn@5, it effectively learns rule patterns when given only partial rules, and in the extreme case with no oracle rules, it still

achieves competitive performance (KL: 0.54, QQ-RMSE: 0.30, MMD: 0.36), still outperforming most baselines (Tab. 2).

Finally, our ablation on reasoning hops $H$ (Tab. 3) shows that the generation quality stabilizes around $H = 4$ when no oracles were provided. Larger $H$ yields only marginal gains, indicating that appropriate multi-hop reasoning improves performance even when no prior rules are available.

| Metrics | Provided Ground Truth Rules | | | | | |
|---|---|---|---|---|---|---|
| | 0 | 1 | 2 | 3 | 4 | 5 (Complete) |
| KL $\downarrow$ | $0.54_{\pm 0.05}$ | $0.50_{\pm 0.03}$ | $0.44_{\pm 0.03}$ | $0.45_{\pm 0.02}$ | $0.43_{\pm 0.02}$ | $0.39_{\pm 0.03}$ |
| QQ-RMSE $\downarrow$ | $0.30_{\pm 0.04}$ | $0.29_{\pm 0.05}$ | $0.30_{\pm 0.03}$ | $0.27_{\pm 0.04}$ | $0.23_{\pm 0.03}$ | $0.22_{\pm 0.05}$ |
| MMD $\downarrow$ | $0.36_{\pm 0.06}$ | $0.38_{\pm 0.05}$ | $0.35_{\pm 0.06}$ | $0.35_{\pm 0.04}$ | $0.32_{\pm 0.03}$ | $0.30_{\pm 0.06}$ |

Table 2: Performance under incomplete rule conditions on Syn@5 dataset. The complete ground-truth rule set consists of 5 rules.

| Metrics | Number of Hops $H$ | | | | |
|---|---|---|---|---|---|
| | 1 | 2 | 3 | 4 | 5 |
| KL $\downarrow$ | $0.54_{\pm 0.05}$ | $0.48_{\pm 0.04}$ | $0.48_{\pm 0.04}$ | $0.42_{\pm 0.02}$ | $0.43_{\pm 0.03}$ |
| QQ-RMSE $\downarrow$ | $0.36_{\pm 0.04}$ | $0.31_{\pm 0.03}$ | $0.32_{\pm 0.02}$ | $0.25_{\pm 0.02}$ | $0.24_{\pm 0.01}$ |
| MMD $\downarrow$ | $0.43_{\pm 0.04}$ | $0.38_{\pm 0.02}$ | $0.37_{\pm 0.03}$ | $0.33_{\pm 0.02}$ | $0.32_{\pm 0.02}$ |

Table 3: Ablation on reasoning hops ($H$) without ground-truth rules on Syn@5 dataset.

**Analysis 3: The Role of Neuro-Symbolic Reasoning Layer and LLM in TPP Sequences Generation** From the results of ablation study in Fig. 2 and Tab. 13 in Appendix. C.9, the neuro-symbolic reasoning layer—which incorporates domain knowledge—significantly improves generation quality. Specifically, augmenting the decoder with inferred global predicate values leads to performance gains of $16.7\%$ in DS, $11\%$ in GPTScore, and $11\%$ in MMD, alongside a substantial reduction in training loss after convergence. Further incorporating the dynamic latent state inference captures instantaneous dependencies among predicates, enabling finer-grained causal reasoning and resulting in additional improvements. Similarly, the multi-round LLM rule refinement module also yields consistent gains across all metrics.

To compare the quality of rules provided by different LLMs, we consider various LLMs from small-scale (Zhang et al., 2024; Team et al., 2024), Opt family (Zhang et al., 2022), and GPT

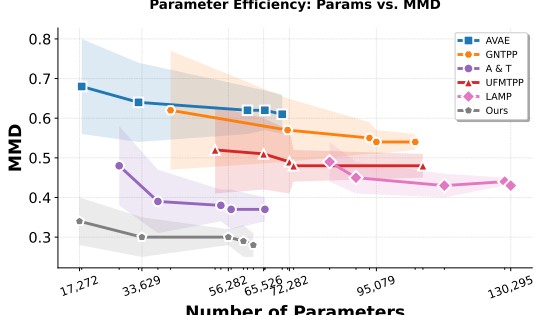

Figure 3: Parameter efficiency comparison of different models on Syn@5 dataset. For each baseline, we varied the model complexity across five settings, from parsimonious to complex architectures. For models such as AVAE, we did not further increase trainable parameter counts beyond a reasonable range, as overly large architectures led to clear overfitting.

family (OpenAI, 2022). Shown in Tab. 10, Appendix. C.8, intriguingly, while the capability for logic rule generation generally improves with model scale—with a particularly notable leap from 1B to 10B parameters— even some smaller LLMs achieve competitive results, demonstrating good adaptability. In Tab. 11-12, Appendix C.8, we observe consistent performance with low variance between different LLM judges, validating that our results are not reliant on the idiosyncrasies of any single LLM.

**Analysis 4: Advantages in Few-Shot and Zero-Shot Generation** Following the experimental pipeline of (Li et al., 2020a), we evaluate model performance in few-shot generation tasks using varying proportions of the training data. As illustrated in Fig. 4, our method demonstrates a consistent advantage under few-shot settings for MIMIC-IV dataset across all data regimes. Even as the amount of training data decreases from $100\%$ to $20\%$, the performance of our approach exhibits only a marginal degradation and consistently surpasses all baselines with $100\%$ training samples.

Furthermore, in Analysis 1, although our model slightly underperforms the A & T baseline on the EPIC-100 dataset when using $100\%$ of the training data, it exhibits strong data efficiency: with reduced data (e.g., $80\%$ or less), our model outperforms all others on EPIC-100 (see Appendix. C.6). These results highlight the applicability of our approach to few-shot generation scenarios, particularly in data-scarce domains such as rare disease data synthesis. We also analyze the scalability and time efficiency of our proposed method, with details can be found in Appendix. C.10.

Remarkably, as presented in Tab 9, Appendix. C.7, by embedding controllable pre-defined domain knowledge into the neuro-symbolic layer, our model achieves zero-shot generation of clinically meaningful sequences that outperform baseline methods.

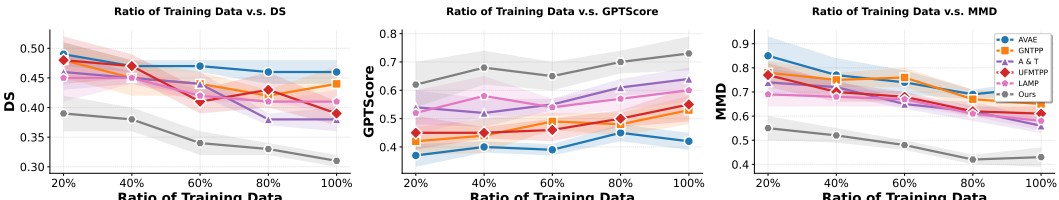

Figure 4: Few-Shot generation comparison for MIMIC-IV dataset, using MMD as evaluation metric. No comparison for SP and QA because they use pre-trained LLM.

**Analysis 5: The Practical and Clinical Significance of Generation under Different Missing Data Patterns.** A common clinical scenario involves handling patients with partially missing data—for instance, when recent laboratory results are available, but historical medical records are absent. It is crucial for a generative model to perform consistently and reliably under such incomplete data conditions. To evaluate this capability, we design an experiment that simulates four types of patient data missingness: initial stage, with missing events in the first $0$–$25\%$, early-middle stage ($25$–$50\%$), late-middle stage ($50$–$75\%$), and final stage ($75$–$100\%$). As shown in Fig. 5, our model achieves good performance and consistently outperforms all baseline methods across most missingness scenarios, except when missing values occur at the initial stages. The performance degradation during initial-stage generation is expected due to the auto-regressive nature of our model, which relies on previous timesteps to inform subsequent events—making it sensitive to missingness early in the sequence. Our experiment empirically quantifies this lower bound and empirically demonstrate how severe this universal issue. Encouragingly, while initial-event generation shows expected performance degradation, our method nevertheless retains good performance, with an MMD of 0.32. Importantly, our model achieves superior performance in later-stage generation, attaining the lowest MMD scores across the other three generation phases. These results demonstrate the robustness and suitability of our approach for generative tasks under realistic conditions of partial observational data.

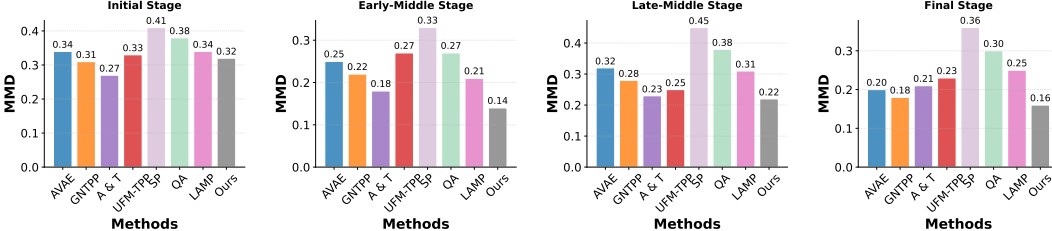

Figure 5: Generation performance comparison with different missing patterns for MIMIC-IV dataset. Define four missing data patterns over the time horizon: initial stage (missing events in the first $0$–$25\%$), early-middle stage ($25$–$50\%$), late-middle stage ($50$–$75\%$), and final stage ($75$–$100\%$).

## 5 CONCLUSION

We introduced NS-VAE-TPP, a neuro-symbolic framework that integrates temporal point processes with rule-guided reasoning for logic-consistent sequence generation. By unifying probabilistic modeling with symbolic constraints, which are provided by LLM, our approach achieves accurate and reliable generation even under data scarcity, demonstrating the promise for robust generation in safety-critical domains.

## REPRODUCIBILITY STATEMENT

We have made extensive efforts to ensure the reproducibility of our results. Details of the computational setup, including hardware configuration and software environment, as well as the choice of hyper-parameters are documented in Appendix. D. The detailed computation process of evaluation metrics are provided in Appendix. C.3. Moreover, a complete description of data preprocessing for the synthetic, semi-synthetic, and real-world dataset is also provided in Appendix. C.1. Upon acceptance, we will release our code to facilitate replication and further research.

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

APPENDIX

## A   ALGORITHM

We provide the overall algorithm pseudocode of the generative process of our proposed model in Alg. 1.

---

**Algorithm 1** Generative process

---

1: **Input:** Initial input sequence $x_0$
2: **Output:** Final output generated sequence $\hat{x} = \{(\hat{t}_i, \hat{m}_i)\}_{i=1}^N$
3: **Sample Once:** at $t = 0$, $\tilde{v}_0^H \sim q(v_0^H \mid x_0)$   ▷ global latent state.
4: **for** each $i = 1, ..., N$ **do**
5:     Encode the history $H_i = \text{Encoder}(x_{<t_i})$.   ▷ current history embedding.
6:     Compute dynamic reasoning state $v_{t_i}^H = \text{ReasoningLayer}(H_i)$.   ▷ dynamic latent state.
7:     Form the decoder input $s_i = [\text{Pool}(H_i); \tilde{v}_0^H; v_{t_i}^H]$.
8:     Sample the next event: $\hat{\Delta} t_i \sim p_\psi(\Delta t_i \mid s_i)$, $\quad \hat{m}_i \sim p_\psi(m_i \mid s_i)$.
9: **end for**
10: **Objective:** Minimize ELBO:

$$\mathcal{L}_{\text{ELBO}} = \mathbb{E}_{\tilde{v}_0^H \sim q(v_0^H \mid x_0)} \left[ \sum_{i=1}^N \log p_\psi(\Delta t_i, m_i \mid s_i) - \text{KL}\big(q(v_0^H \mid x_0) \,\|\, p(v_0)\big) \right].$$

---

## B   PROMPT DESIGN

**Prompt for Querying Initial Rule Set**   The prompt for querying initial rule set is given by Fig. 6.

**Prompt for Rule Set Refinement**   The prompt for refine initial/current rule set is given by Fig. 7.

**Selection of Representative Sequences**   For initializing symbolic rules via LLMs, we use a small set of representative sequences from the training data. These sequences are chosen to capture diverse patterns in the event logs, ensuring they reflect common domain scenarios. In our experiments in Tab. 4, we found that 3–5 sequences are sufficient to elicit meaningful rules from the LLM. See the table below, for MIMIC-IV dataset, 3–5 examples are generally sufficient to provide meaningful context and elicit high-quality rules from the LLM, while increasing this number (e.g., to 9) yields only marginal improvements.

| Metrics | Number of Representative Sequences | | | | |
|---|---|---|---|---|---|
| | **1** | **3** | **5** | **7** | **9** |
| DS ↓ | $0.38_{\pm 0.03}$ | $0.32_{\pm 0.02}$ | $0.31_{\pm 0.01}$ | $0.30_{\pm 0.02}$ | $0.30_{\pm 0.01}$ |
| GPTScore ↑ | $0.67_{\pm 0.07}$ | $0.73_{\pm 0.06}$ | $0.73_{\pm 0.06}$ | $0.73_{\pm 0.04}$ | $0.73_{\pm 0.04}$ |
| MMD ↓ | $0.49_{\pm 0.03}$ | $0.45_{\pm 0.04}$ | $0.43_{\pm 0.04}$ | $0.43_{\pm 0.03}$ | $0.41_{\pm 0.01}$ |

Table 4: Selection of representative samples for MIMIC-IV dataset. Our current setting is 5 representative sequences.

## C   EXPERIMENTAL DETAILS

### C.1   DATASET

We provide the overall dataset statistics in Tab. 5. And we also provide the details of pre-processing as followings:

**Synthetic Datasets**   *(i)* **Syn@5**: We create sequences consisting of 5 event predicates, sampled from a prespecified Temporal Logic Point Process (TLPP) (Li et al., 2020b). Specifically, we employ multiple pre-defined logic rules along with their weights to construct the intensity function, and then apply thinning algorithms (Ogata, 1981) to generate new events. To evaluate the scalability of the proposed model, we have created five distinct groups, with the sample size varying from 1000 to 9000 (see Appendix. C.10). In the experiment of main context, we use the dataset with 5000

---

## LLM Prompt: Initial Logic Rule Generation

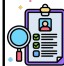

**Background:**
You are a logic reasoning assistant tasked with extracting high-level predicates and logic rules based on observed temporal point process events. Your goal is to identify high-level predicates based on provided event types and logic rules that can capture patient physical condition built on extracted predicates to identify the occurrence patterns in temporal event sequences. You are given a set of event types and some of the examples of sequences. The event type set consists of 38 different clinical occurred events with clinical meaning. And event sequences records in the format of series of events, e.g. ("event time" @ "event time").

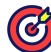

**Your Goal:**
Extract high-level predicates and generate reasonable and clinical meaningful logic rules based on given clinical event types.

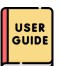

**Instructions:**
1. Think aloud: Propose reasonable number of high-level predicates.
1. Think aloud: Propose 3–5 candidate logic rules.
2. Explain briefly why each rule might be effective.

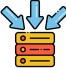

**Example Input:**
Event type set: [abnormal_sysbp, abnormal_spo2_sao2, abnormal_cvp, abnormal_svr, abnormal_potassium_meql, abnormal_sodium, abnormal_chloride, abnormal_bun, abnormal_creatinine, abnormal_crp, abnormal_rbc_count, abnormal_wbc_count, abnormal_arterial_ph, abnormal_arterial_be, abnormal_arterial_lactate, abnormal_hco, abnormal_svo2_scvo2, normal_sysbp, normal_spo2_sao2, normal_cvp, normal_svr, normal_potassium_meql, normal_sodium, normal_chloride, normal_bun, normal_creatinine, normal_crp, normal_rbc_count, normal_wbc_count, normal_arterial_ph, normal_arterial_be, normal_arterial_lactate, normal_hco3, normal_svo2_scvo2, low_urine, intravenous_fluids, use_drugs, survival].

**Event Sequences Examples:**
Sequence 1: [abnormal_potassium_meql @ 0.49, abnormal_sodium @ 1.39, normal_arterial_ph @ 1.78, ...]
Sequence 2: [abnormal_arterial_be @ 1.58, normal_arterial_ph @ 1.58, abnormal_sysbp @ 1.62, ...]
Sequence 3: [survival @ 0.00, abnormal_potassium_meql @ 0.01, normal_arterial_lactate @ 0.01, ...]

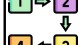

**Your Tasks:**
Extract high-level predicates and generate symbolic rules based on these predicates. For each rule, include 1–2 lines of explanation.

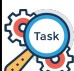

**Output Example:**
Extracted predicates: A, B, C, ...
Generated rules: IF A AND B AND C AND D, THEN E, ...
Explanation: Because XXX, XXX...

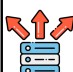

Figure 6: Prompt for querying initial rule set.

864
865
866
867
868
869
870
871
872
873
874
875
876
877
878
879
880
881
882
883
884
885
886
887
888
889
890
891
892
893
894
895
896
897
898
899
900
901
902
903
904

---

## LLM Prompt: Initial Logic Rule Generation

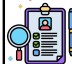

**Background:**
You are a logic reasoning assistant tasked with extracting more logic rules except for the given logic rules. You are also given some observed temporal point process events which cannot be explained well by the already provided logic rules. Your goal is to identify more different logic rules that can capture patient physical condition built on previously extracted predicates to identify the occurrence patterns in the provided temporal event sequences. You are given some of the sequences that cannot be explained well by the provided logic rules. And event sequences records in the format of series of events, e.g. ("event time" @ "event time"). Please note that the rules should build on previous extracted high-level concepts

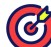

**Your Goal:**
Generate extra reasonable and clinical meaningful logic rules based on previously extracted high-level predicate and provided sequences which cannot be explained well by already provided logic rules.

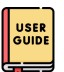

**Instructions:**
1. Think aloud: Propose 1–2 more candidate logic rules.
2. Explain briefly why each rule might be effective.

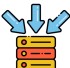

**Event Sequences Examples that cannot Be Explained Well by the Provided Logic Rules:**
Sequence 1: [abnormal_sodium @ 0.01, abnormal_chloride @ 0.01, abnormal_bun @ 0.01, ...]
Sequence 2: [abnormal_potassium_meql @ 0.05, normal_arterial_ph @ 0.05, abnormal_spo2_sao2 @ 0.10, ...]
Sequence 3: [survival @ 0.00, normal_arterial_ph @ 1.81, normal_arterial_be @ 1.81, ...]

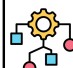

**Provided Logic Rules:**
Rule-1: IF Cardiovascular Instability AND Electrolyte Imbalance THEN Renal Dysfunction
Rule-2: IF Respiratory Dysfunction AND Acid–Base Disturbance THEN Cardiovascular Instability
Rule-3: IF Electrolyte Imbalance AND Renal Dysfunction THEN Inflammatory / Immune Response
Rule-4: IF Cardiovascular Instability AND Treatment Intervention THEN Survival
Rule-5: IF Respiratory Dysfunction AND Inflammatory / Immune Response THEN Acid–Base Disturbance

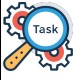

**Your Tasks:**
Generate symbolic rules that are different with the given logic rules, which can best explain the given temporal point process sequences. For each rule, include 1–2 lines of explanation.

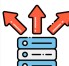

**Output Example:**
Generated rules: IF A AND B AND C AND D, THEN E.
Explanation: Because XXX, XXX...

Figure 7: Prompt for refine initial/current rule set.

905
906
907
908
909
910
911
912
913
914
915
916
917

samples and an average sequence length of 20.56 events. *(ii)* **Syn@10**: We make the synthetic setting more challenging by extending the number of predicates to 10 and further increasing the number of underlying rules, thereby enlarging the complexity of the underlying logical space. This setup enables us to assess the model's capacity to cope with richer and more intricate scenarios, as well as to evaluate its robustness and generalization ability under more demanding conditions. In our main experiments, we employ a dataset comprising 5000 samples with an average sequence length of 38.71 events per sample.

**Semi-Synthetic Datasets** *(iii)* **LogiCity** (Li et al., 2024a): LogiCity is an urban-scale multi-agent simulator grounded in customizable first-order logic (FOL). It models diverse urban entities through semantic and spatial concepts, which are used to formulate FOL rules that govern agent behaviors. In our initial case study, we simulate 8 distinct agents (6 car agents and 2 pedestrian agent), each characterized by predicates such as `IsAtIntersection`, `Stop`, and `CollidingClose`, resulting in a total of 24 predicates. We generate 500 sequences to facilitate evaluation of interpretable rule-based reasoning for generation tasks in dynamic environments. This platform serves as a testbed for neuro-symbolic methods in complex, interactive scenarios.

**Real-World Datasets** *(iv)* **MIMIC-IV** (Johnson et al., 2020): MIMIC-IV is a publicly available electronic health record dataset comprising patients admitted to the intensive care unit (ICU). We focus on those diagnosed with sepsis—a life-threatening condition and major cause of ICU mortality (Saria, 2018). We extract 2023 sequences with an average event count 33.10. Predicates are primarily categorized into two types: medication-related (e.g., `Use Drugs` or `Intravenous Fluids`) and clinical measurement-related (e.g., `Low Urine`, `Abnormal SpO2SaO2`, or `Abnormal ArterialPH`). These predicates support structured representation and logic-based reasoning for clinical event sequences. *(v)* **EPIC-100** (Damen et al., 2020): This dataset is derived from a large-scale egocentric vision corpus, consisting of unstructured audio-visual recordings captured in natural home environments. It focuses on daily kitchen activities observed over multiple days. From the annotated action sequences, we extract temporal event histories comprising only cooking verbs—intentionally omitting the interacted objects to emphasize verb-centric reasoning. The dataset includes 68 verbs, such as `Put-In`, `Rinse`, `Put-On`, `Pour`, `Stir`, `Peel`, `Chop`, and `Slice`, among others. In total, the dataset contains 8656 sequences, with an average length of 34.84 events per sequence. *(vi)* **IKEA ASM** (Ben-Shabat et al., 2021): This dataset comprises 371 unique assembly configurations of four furniture types—side table, coffee table, TV bench, and drawer—each available in three colors: white, oak, and black. We focus specifically on the TV bench assembly task, which involves 18 action predicates such as `Pick Up Leg`, `Tighten Leg`, and `Attach Shelf to Table`. The subset includes 91 sequences with an average length of 24.46 events per sequence.

| Category | Dataset | Statistics | | | |
|---|---|---|---|---|---|
| | | # Predicates | # Sequences | Events Average Length | Time Horizon |
| **Synthetic** | **Syn@5** | 5 | 5000 | 20.56 | 20 |
| | **Syn@10** | 10 | 5000 | 38.71 | 20 |
| **Semi-Synthetic** | **LogiCity** | 24 | 500 | 56.25 | 100 |
| **Real-World** | **MIMIC-IV** | 38 | 2023 | 33.10 | 118.42 |
| | **Epic-100** | 68 | 8656 | 34.84 | 1585 |
| | **IKEA ASM** | 18 | 91 | 24.46 | 5955 |

Table 5: Dataset statistics. For Epic-Kitchen-100 and IKEA ASM datasets, we scale the time horizon to 0-200.

## C.2 BASELINES

We choose state-of-the-art baselines considering two different fields:

**Neural Temporal Point Process Models** *(i)* **AVAE** (Mehrasa et al., 2019): The model is a recurrent variational auto-encoder designed for modeling asynchronous action sequences. At each time step, the model utilizes the history of actions and inter-arrival times to generate a distribution over latent variables. A sample from this distribution is then decoded into probability distributions for

the inter-arrival time and action label of the next action. To address the limitations of using a fixed prior in the traditional VAE framework, this model incorporates a prior net that enhances the learning process. *(ii)* **GNTPP** (Lin et al., 2022): The model is a comprehensive generative framework for neural temporal point process modeling. It leverages deep generative models as probabilistic decoders, including the temporal conditional diffusion denoising model, temporal conditional VAE, temporal conditional GAN, temporal conditional continuous normalizing flow, and temporal conditional noise score network models. The diverse combinations of encoders and decoders make the GNTPP highly flexible in approximating the target distribution of event occurrence times. For the encoder, the model incorporates both RNN-based approaches and self-attention mechanisms. In our experiments, we select the revised attentive history encoder and the VAE probabilistic decoder. *(iii)* **A & T (Add-and-Thin)** (Lüdke et al., 2023): A non-autoregressive probabilistic denoising diffusion model for TPPs capable of generating complete event sequences in parallel. It offers flexible handling of variable event counts and continuous time, overcoming common limitations of auto-regressive models. *(iv)* **UFM-TPP** (Shou, 2025): It proposes a unified flow-matching framework for jointly modeling inter-event times and event types (marks), utilizing both continuous and discrete flow matching to achieve integrated generation. This approach enables the model to directly map from simple noise distributions—such as exponential for times and uniform for types—to realistic event sequences, without relying on step-by-step recursion.

**LLM-Based Models** *(v)* **SP (Simple-Prompt LLM Generation)** (Si et al., 2022): For simple-prompt LLM generation, most existing works on prompt for event generation/extraction aim for teh event extraction (identifying event triggers and parameters from text); and story generation (generating narrative events (who did what)). In these works, "event" refers to events in the linguistic sense, rather than strictly stochastic modeling of temporal processes. Therefore, they are not directly equivalent to "TPP sequence generation". However, their methodologies can still inspire attempts at TPP generation: Prompt-as-sampler/Template prompting: Like the Simple Event Extraction Framework, this approach dispenses with complex probabilistic modeling and directly uses prompts to ask the LLM to output "$(t_1, \texttt{event type}_1), (t_2, \texttt{event type}_2)...$". *(vi)* **QA (QA-based LLM Generation)** (Castricato et al., 2021): For QA-based Generation, according to the idea of EDGAR (Castricato et al., 2021), the model can recursively generate the next event and its time through question-answering prompts. *(vii)* **LAMP** (Shi et al., 2023): A framework for TPPs that integrates a LLM. In particular, the LLM performs abductive reasoning to support the event sequence model. Guided by a few expert-annotated demonstrations, the LLM learns to suggest plausible causes for each candidate event. A search module then identifies previous events that align with these causes, and a scoring function evaluates whether the retrieved events could indeed lead to the proposed event. To adapt to TPP generation tasks, the original LAMP's proposer—which previously suggested candidate future events—has been replaced with an unconditional or conditional full-sequence candidate generator that produces a complete segment in a single pass. The process retains the LLM's abductive reasoning and re-ranking steps to filter and reorder candidates, ultimately outputting the final sample.

## C.3 COMPUTATION OF METRICS

We consider two aspects:

**In-Distribution Generation** For fine-tuned generation, we can directly compare the generated sequences with the input sequences, since both are expected to follow similar patterns. Therefore, we can use following metrics to evaluate the quality of the generated sequences:

*(i)* **KL Divergence**: The synthetic datasets are generated by the temporal logic point process (TLPP). Therefore, we know the ground-truth logic rules and the corresponding rule weights, which allows us to compute the likelihood of each sequence. We first define the input sequence set and the output sequence set as $\boldsymbol{X} = \{\boldsymbol{X_i}\}_{i=1}^n$ and $\boldsymbol{Y} = \{\boldsymbol{Y_j}\}_{j=1}^n$. A sequence $\boldsymbol{S} = \{(t_l, a_l)\}_{l=1}^{N(\boldsymbol{S})}$ on the time horizon $[0, T]$. The event type $a_l \in \{1, ..., K\}$. Then the TLPP intensity for each event type $k$ can be computed as

$$\lambda_k(t|\mathcal{H}_t) = g(\eta_k(t)), \text{ where } \eta_k(t) = b_k + \sum_{f=1}^F w_{f,k}\phi_{f,k}(\mathcal{H}_t, t)$$

where $g(\cdot) > 0$, e.g., $g(x) = \exp(x)$. Total intensity can be written as

$$\Lambda(t) = \sum_{k=1}^K \lambda_k(t|\mathcal{H}_t)$$

Then compute the log-likelihood for each sequence. For an arbitrary sequence $\boldsymbol{S}$

$$l(\boldsymbol{S}) = \sum_{l=1}^{N(\boldsymbol{S})} \log \lambda_{a_l}(t_l|\mathcal{H}_{t_l}) - \int_0^T \Lambda(t)dt$$

To remove scale effects from sequence length or time window, we compute the per-event normalization and the per-time normalization.

$$l_{\text{per-event}}(\boldsymbol{S}) = \frac{l(\boldsymbol{S})}{N(\boldsymbol{S})} \text{ (if } N(\boldsymbol{S}) > 0), \ l_{\text{per-time}}(\boldsymbol{S}) = \frac{l(\boldsymbol{S})}{T}$$

Take standardization $l_{\text{per-event}}(\boldsymbol{S})/l_{\text{per-time}}(\boldsymbol{S})$ and collect vectors
$$\boldsymbol{l_X} = \{l(\boldsymbol{X_1}), ..., l(\boldsymbol{X_n})\}, \ \boldsymbol{l_Y} = \{l(\boldsymbol{Y_1}), ..., l(\boldsymbol{Y_n})\}$$
Estimate the log-likelihood density of two groups using kernel density estimation (KDE)

$$\hat{p}(x) = \frac{1}{nh} \sum_{i=1}^n k(\frac{x - l_X^{(i)}}{h}), \ \hat{q}(y) = \frac{1}{nh} \sum_{j=1}^n k(\frac{y - l_Y^{(j)}}{h})$$

Then the KL divergence between the two log-likelihood distribution are given by

$$\hat{\text{KL}}(\hat{p}||\hat{q}) = \frac{1}{n} \sum_{i=1}^n \log \frac{\hat{p}(l(\boldsymbol{X_i}))}{\hat{p}(l(\boldsymbol{Y_i}))}$$

*(ii)* **QQ-RMSE** (Xiao et al., 2017): Similarly we can get the KDE or histogram of $\boldsymbol{l_x}$ and $\boldsymbol{l_Y}$ on same axes. Then we compute the empirical quantiles $Q_X(\alpha), Q_y(\alpha)$ for $\alpha \in (0, 1)$. plot pairs $(Q_X(\alpha), Q_Y(\alpha))$. If distributions match, points lie on $y = x$. Implement discrete sampling by using $\alpha_k = \frac{k}{m+1}, k = 1, ..., m$. Then we can quantify the QQ deviation

$$\text{QQ-RMSE} = \sqrt{\frac{1}{m} \sum_{k=1}^m (Q_X(\alpha_k) - Q_Y(\alpha_k))^2}$$

*(iii)* **MMD (Maximum Mean Discrepancy)** (Shchur et al., 2020): We computed MMD (Gretton et al., 2012) between the original sequences and the generated sequences after training. MMD quantifies the dissimilarity between the true data distribution $p^*(\boldsymbol{t})$ and the learned density $p(\boldsymbol{t})$ — lower is better. Now we provide the computation process:
Following above definition, the input sequence set is denoted as
$$\boldsymbol{X} = \{\boldsymbol{X_1}, \boldsymbol{X_2}, ..., \boldsymbol{X_n}\}, \ \text{each } \boldsymbol{x_i} = [(t_{i1}, a_{i1}), (t_{i2}, a_{i2}), ..., (t_{iL_i}, a_{iL_i})]$$
And the output sequence set is denoted as
$$\boldsymbol{Y} = \{\boldsymbol{Y_1}, \boldsymbol{Y_2}, ..., \boldsymbol{Y_n}\}, \ \text{each } \boldsymbol{y_j} = \left[(t_{j1}, a_{j1}), (t_{j2}, a_{j2}), ..., (t_{jL_j}, a_{jL_j})\right]$$
For each sequence in input set, we build sequence embedding
$$\boldsymbol{e_{ik}} = [\psi(t_{ik}); \text{emb}(a_{ik})] \in \mathbb{R}^d$$
where $\psi(\cdot)$ is the positional embedding for event time and $\text{emb}(\cdot)$ is the one-hot embedding for event type. Take the average of the entire sequence, we obtain

$$\tilde{\boldsymbol{X_i}} = \frac{1}{L_i} \sum_{k=1}^{L_i} \boldsymbol{e_{ik}}, \ \tilde{\boldsymbol{Y_j}} = \frac{1}{L_j} \sum_{k=1}^{L_j} \boldsymbol{e_{jk}}$$

Then, we define the kernel function (RBF kernel, i.e., Gaussian kernel)
$$k(\boldsymbol{A}, \boldsymbol{B}) = \exp\left(-\frac{||\boldsymbol{A} - \boldsymbol{B}||^2}{2\sigma^2}\right)$$
Finally, we can obtain unbiased MMD estimation

$$\text{MMD}^2(\boldsymbol{X}, \boldsymbol{Y}) = \frac{1}{n(n-1)} \sum_{i \neq j} k(\tilde{\boldsymbol{X_i}}, \tilde{\boldsymbol{X_j}}) + \frac{1}{n(n-1)} \sum_{i \neq j} k(\tilde{\boldsymbol{Y_i}}, \tilde{\boldsymbol{Y_j}}) - \frac{2}{n^2} \sum_{i=1}^n \sum_{j=1}^n k(\tilde{\boldsymbol{X_i}}, \tilde{\boldsymbol{Y_j}})$$

*(iv)* **DS (Discriminator Scores)** (Desai et al., 2021): The generated sequences are used to train the post-hoc sequence classifi-cation models (by optimizing a 2-layer LSTM) to distinguish between sequences from the original and generated sequences. First, each original sequence is labeled "*Real*", and each generated sequence is labeled "*Not Real*". Then, an off-the-shelf classifier is trained to distinguish between the two classes as a standard supervised task. Therefore, we obtain the discrimination scores (`accuracy` - 0.5). A score close to 0 is better, indicating the generated data is hard to distinguish from original data.

*(v)* **GPTScore** (Fu et al., 2023): A new framework that evaluates event sequence texts with generative pre-training models like GPT-3. It assumes that a generative pre-training model will assign a higher probability of high-quality generated event sequence text following a given instruction and context. Note that the metrics KL and QQ-RMSE require ground truth from the temporal logic point process model. Consequently, they are only applicable to synthetic datasets.

**Rule-Conditioned (Out-of-Distribution) Generation** However, in the case of zero-shot generation, the process is guided by modifying the underlying logic rules. As a result, the generated sequences may exhibit patterns that differ substantially from previously observed data, making direct comparison with input sequences less appropriate. To address this, we turn to LLMs as judges, evaluating the generated sequences along two dimensions: whether they adhere to the specified logic rules, and whether they align with realistic, domain-specific meaning.

*(vi)* **R-Score (Rule-based GPTScore)**: Measures how well the generated sequence adheres to the predefined logic rules, with the assessment performed by GPT. Higher scores indicate that the sequence better follows the specified logic rules.

*(vii)* **C-Score (Clinical GPTScore)**: Evaluates the overall plausibility and acceptability of the generated sequence, with the GPT assessing factors such as event order, timing, and domain knowledge. Higher scores indicate that the sequence is reasonable, meaningful, and consistent with expected patterns.

## C.4 VISUALIZATION OF INPUT AND OUTPUT SEQUENCE COMPARISON

In the main text, we report quantitative analyses across multiple evaluation metrics. To complement these results, Fig. 8 presents a representative input–output comparison from the MIMIC-IV dataset after training convergence. The generated sequences basically align with the observed sequences in both event frequency and temporal structure, indicating that our model not only reconstructs the training data effectively but also achieves good-quality sequence generation.

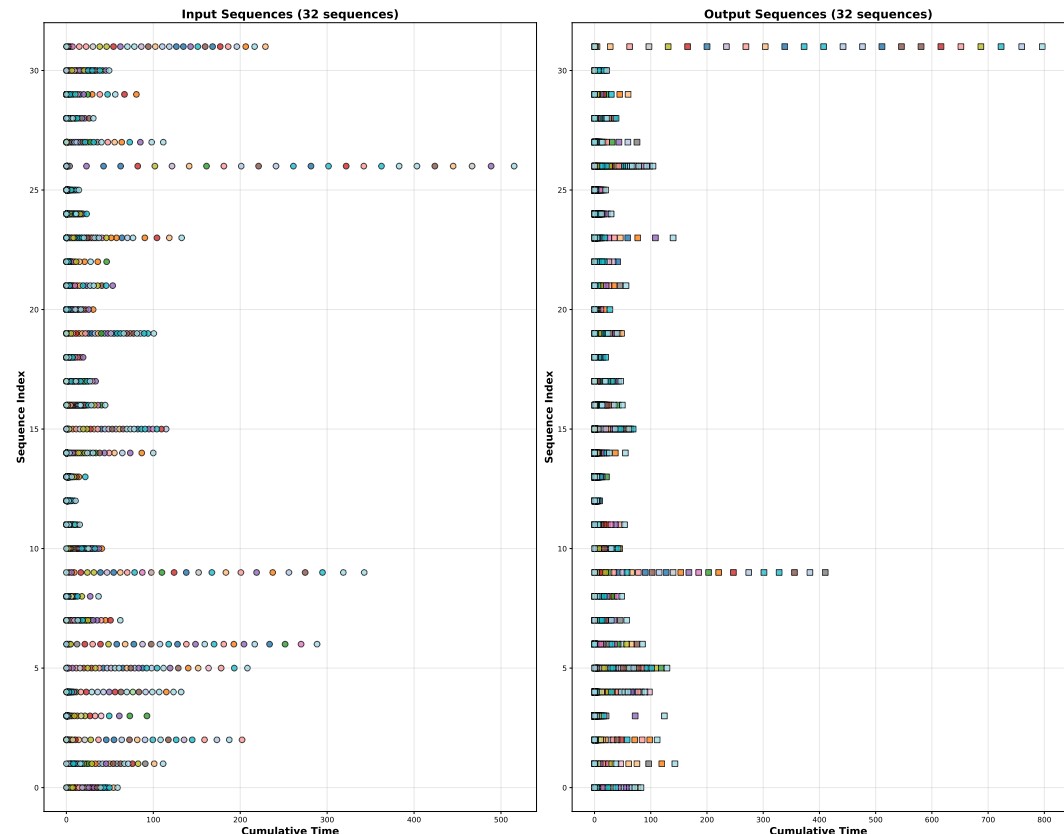

Figure 8: Comparison of model inputs and generated outputs on a representative batch from the MIMIC-IV dataset at convergence.

## C.5 REPRESENTATIVE QUERIED RULES WITH REAL-WORLD SIGNIFICANCE ON REAL-WORLD DATASET

We present part of representative logic rules queried from LLM for MIMIC-IV, EPIC-Kitchen-100, IKEA ASM real-world datasets in Tab. 6, Tab. 7, and Tab. 8. These logic rules are well-aligned with the domain common sense and realistic constraints of the datasets, and are therefore integrated as domain knowledge into the neuro-symbolic reasoning layer to support structured and interpretable inference.

---

**Part of Rules Queried from LLM for MIMIC-IV dataset**

---

**Rule-1**: Renal Dysfunction ← Cardiovascular Instability ∧ Electrolyte Imbalance
**Rule-2**: Cardiovascular Instability ← Respiratory Dysfunction ∧ Acid-Base Disturbance
**Rule-3**: Inflammatory / Immune Response ← Electrolyte Imbalance ∧ Renal Dysfunction
**Rule-4**: Survival ← Cardiovascular Instability ∧ Treatment Intervention
**Rule-5**: Acid-Base Disturbance ← Respiratory Dysfunction ∧ Inflammatory / Immune Response
...... ......

---

Table 6: Part of logic rules queried from LLM for MIMIC-IV dataset.

---

**Part of Rules Queried from LLM for Epic-Kitchen-100 dataset**

---

**Rule-1**: Put-Into (Something) ← Open (Something) ∧ Take (Something) ∧ Cut (Something)
**Rule-2**: Pour-Into (Something) ← Turn-On (Something) ∧ Put (Something)
**Rule-3**: Turn-Off (Something) ← Turn-On (Something) ∧ Wash (Something)
**Rule-4**: Chop (Something) ← Take (Something) ∧ Cut (Something) ∧ Put-Down (Something)
**Rule-5**: Grab (Something) ← Open (Something) ∧ Take (Something) ∧ Close (Something)
        ∧ Put-Down (Something)
...... ......

---

Table 7: Part of logic rules queried from LLM for Epic-Kitchen-100 dataset.

---

**Part of Rules Queried from LLM for IKEA ASM dataset**

---

**Rule-1**: Tighten Leg ← Pick Up Leg ∧ Align Leg Screw with Table Thread ∧ Spin Leg
**Rule-2**: Tighten Leg ← Align Leg Screw with Table Thread ∧ Spin Leg
**Rule-3**: Rotate Table ← Tighten Leg
**Rule-4**: Flip Table ← Tighten Leg
**Rule-5**: Attach Shelf to Table ← Flip Table ∧ Pick Up Shelf
**Rule-6**: Pick Up Leg ← Flip Table Top
...... ......

---

Table 8: Part of logic rules queried from LLM for IKEA ASM dataset.

## C.6 FEW-SHOT GENERATION FOR EPIC-KITCHEN-100 DATASET

As presented in Analysis 1 of the main text, our model exhibits slightly lower performance than the A & T baseline on the EPIC-Kitchen-100 dataset under full data ($100\%$ training) in terms of DS and GPTScore. However, it demonstrates stronger data efficiency when less training data is available: with $80\%$ or fewer samples, our model outperforms all baselines on EPIC-Kitchen-100. In the extreme low-data regime using only $20\%$ of the training data, our method substantially surpasses other approaches in sequence generation quality. Specifically, with $20\%$ data, our model achieves DS = 0.42, GPTScore = 0.53, and MMD = 1.25. In comparison, the second-best results are achieved by GNTPP (DS = 0.46, GPTScore = 0.41) and by LAMP (MMD = 1.33). These findings underscore the suitability of our approach for few-shot generation, especially in data-scarce applications such as synthesizing rare disease data.

## C.7 ZERO-SHOT GENERATION FOR MIMIC-IV DATASET

Encouragingly, the integration of domain knowledge into the neuro-symbolic reasoning layer enables zero-shot generation capability in our model. By pre-defining desired logic rules that the generated data must satisfy, our well-trained model can produce sequence data conforming to these rules in a zero-shot manner. As shown in Tab. 9, we designed logic rules focusing on two aspects: *drug effects*

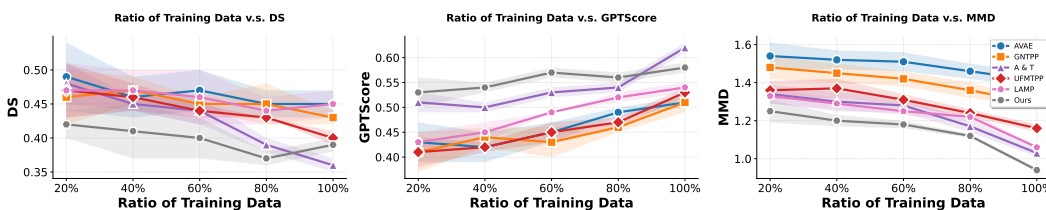

Figure 9: Few-Shot generation comparison for Epic-Kitchen-100 dataset, using MMD as evaluation metric. No comparison for SP and QA because they use pre-trained LLM.

*(i.e., how medications influence clinical measurements)* and *lab test interactions (i.e., correlations among patient physiological indicators)*. Using an LLM as a judge, our method outperforms zero-shot baselines such as SP and QA across both R-Score and C-Score, demonstrating stronger generalization under novel rule constraints.

| Category | Rule Set | Metrics | |
|---|---|---|---|
| | | R-Score ↑ | C-Score ↑ |
| **Drugs** | **Rule-1**: `Treatment Intervention ← Cardiovascular Instability ∧ Respiratory Dysfunction`
**Rule-2**: `Treatment Intervention ← Electrolyte Imbalance ∧ Acid-Base Disturbance`
**Rule-3**: `Survival ← Treatment Intervention ∧ Renal Dysfunction`
...... ...... | 0.67 | 0.70 |
| **Lab Tests** | **Rule-1**: `Acid-Base Disturbance ← ∧ Cardiovascular Instability`
**Rule-2**: `Renal Dysfunction ← Respiratory Dysfunction ∧ Inflammatory/Immune Response ∧`
**Rule-3**: `Cardiovascular Instability ← Electrolyte Imbalance ∧ Acid-Base Disturbance`
...... ...... | 0.63 | 0.68 |

*Note*: **Drug**: For baseline **SP**: **R-Score**: 0.43, **C-Score**: 0.45. For baseline **QA**: **R-Score**: 0.46, **C-Score**: 0.45.
**Lab Tests**: For baseline **SP**: **R-Score**: 0.51, **C-Score**: 0.55. For baseline **QA**: **R-Score**: 0.58, **C-Score**: 0.60.

Table 9: Zero-shot generation for MIMIC-IV dataset. We query LLM for pre-defined logic rule in two categories: drug effects (i.e., how medications influence clinical measurements), and lab test interactions (i.e., correlations among patient physiological indicators).

## C.8 COMPARE DIFFERENT LLMS

**Compare Different LLMs as Symbolic Prior Bank**   To compare the quality of rules provided by different LLMs, we consider TinyLlama-1.1B-Chat-v1.0 (Zhang et al., 2024), Gemma-2-2B-IT (Team et al., 2024), Opt family (Zhang et al., 2022) (Opt-125M, Opt-1.5B), and GPT family (OpenAI, 2022): (GPT-3.5, GPT-4o). As shown in Tab. 10, intriguingly, while the capability for logic rule generation generally improves with model scale—with a particularly notable leap from 1B to 10B parameters—we observe that even some smaller LLMs achieve competitive results. For instance, using OPT-1.5B as the rule generator still yields performance on MIMIC-IV that substantially outperforms the baselines in Tab. 1. This demonstrates the compatibility of our approach with a range of pre-trained language models.

| Methods | Metrics | | | |
|---|---|---|---|---|
| | Loss ↓ | DS ↓ | GPTScore ↑ | MMD ↓ |
| **TinyLlama** | $162.94_{\pm 21}$ | $0.39_{\pm 0.00}$ | $0.65_{\pm 0.00}$ | $0.50_{\pm 0.02}$ |
| **Gemma-2** | $162.28_{\pm 27}$ | $0.34_{\pm 0.00}$ | $0.68_{\pm 0.04}$ | $0.47_{\pm 0.02}$ |
| **Opt-125M** | $163.21_{\pm 25}$ | $0.38_{\pm 0.01}$ | $0.63_{\pm 0.02}$ | $0.54_{\pm 0.02}$ |
| **Opt-1.5B** | $162.45_{\pm 24}$ | $0.33_{\pm 0.01}$ | $0.66_{\pm 0.02}$ | $0.48_{\pm 0.01}$ |
| **GPT-3.5** | $162.22_{\pm 16}$ | $0.33_{\pm 0.01}$ | $0.70_{\pm 0.03}$ | $0.46_{\pm 0.03}$ |
| **GPT-4o** | $161.33_{\pm 0.22}$ | $0.31_{\pm 0.01}$ | $0.73_{\pm 0.06}$ | $0.43_{\pm 0.04}$ |

Table 10: Compare the effect of rules generated by different LLMs on MIMIC-IV dataset.

**Compare Different LLMs as Judges**   We employed multiple LLM judges (including GPT-4o and smaller open-source models) to reduce dependency on a single model and reported results with means

and variances. Across both in-distribution (Tab. 11) and zero-shot generation (Tab. 12) evaluations on MIMIC-IV dataset, we observe consistent performance with low variance between different LLMs. This indicates strong agreement in quality assessment across model scales, validating that our results are not reliant on the idiosyncrasies of any single LLM.

| Metrics | LLMs used | | | | |
|---|---|---|---|---|---|
| | GPT-4o | GPT-4o + Opt-1.5B | GPT-4o + Opt-1.5B + Opt-125M | GPT-4o + Opt-1.5B + Opt-125M + Gemma-2 | GPT-4o + Opt-1.5B + Opt-125M + Gemma-2 + TinyLlama |
| GPTScore ↑ | $0.72_{\pm 0.00}$ | $0.73_{\pm 0.01}$ | $0.74_{\pm 0.01}$ | $0.72_{\pm 0.02}$ | $0.72_{\pm 0.02}$ |

Table 11: Robustness evaluation with multiple LLM judges on MIMIC-IV (GPTScore).

| Metrics | LLMs used | | | | |
|---|---|---|---|---|---|
| | GPT-4o | GPT-4o + Opt-1.5B | GPT-4o + Opt-1.5B + Opt-125M | GPT-4o + Opt-1.5B + Opt-125M + Gemma-2 | GPT-4o + Opt-1.5B + Opt-125M + Gemma-2 + TinyLlama |
| R-Score ↑ | $0.67_{\pm 0.00}$ | $0.69_{\pm 0.02}$ | $0.69_{\pm 0.02}$ | $0.68_{\pm 0.03}$ | $0.68_{\pm 0.02}$ |
| C-Score ↑ | $0.69_{\pm 0.00}$ | $0.71_{\pm 0.02}$ | $0.70_{\pm 0.02}$ | $0.71_{\pm 0.02}$ | $0.71_{\pm 0.02}$ |

Table 12: Zero-shot generation robustness on MIMIC-IV with multiple LLM judges (R/C-Scores). The pre-defined rules focus on drug effects (i.e., how medications influence clinical measurements).

## C.9 ABLATION STUDY

| Ablation | | | | Metrics | | | |
|---|---|---|---|---|---|---|---|
| Eq. (4): $\mathrm{Pool}(\boldsymbol{H}_t)$ | Eq. (13): $\tilde{\boldsymbol{v}}_0^H$ | Eq. (14): $\boldsymbol{v}_t^H$ | Rule Refine | Loss ↓ | DS ↓ | GPTScore ↑ | MMD ↓ |
| ✓ | ✗ | ✗ | ✗ | $173.08_{\pm 0.16}$ | $0.48_{\pm 0.03}$ | $0.62_{\pm 0.03}$ | $0.55_{\pm 0.04}$ |
| ✓ | ✓ | ✗ | ✗ | $168.51_{\pm 0.11}$ | $0.40_{\pm 0.03}$ | $0.69_{\pm 0.03}$ | $0.49_{\pm 0.04}$ |
| ✓ | ✓ | ✓ | ✗ | $162.34_{\pm 0.28}$ | $0.36_{\pm 0.02}$ | $0.70_{\pm 0.02}$ | $0.48_{\pm 0.03}$ |
| ✓ | ✓ | ✓ | ✓ | $\mathbf{161.33_{\pm 0.22}}$ | $\mathbf{0.31_{\pm 0.01}}$ | $\mathbf{0.73_{\pm 0.06}}$ | $\mathbf{0.43_{\pm 0.04}}$ |

Table 13: Ablation study on MIMIC-IV. We ablate the following modules: *(i)* $\mathrm{Pool}(\boldsymbol{H}_t)$, Eq. (4): history embedding from encoder. *(ii)* $\tilde{\boldsymbol{v}}_0^H$, Eq. (13): global latent state from neuro-symbolic reasoning layer. *(iii)* $\boldsymbol{v}_t^H$, Eq. (14): dynamic latent state for each time step from the generated history. *(iv)* Rule Refine: multi-round LLM rule refinement. The Loss v.s. Epoch curves are shown in Fig. 2.

From the results of ablation study in Fig. 2 and Tab. 13, the neuro-symbolic reasoning layer—which incorporates domain knowledge—significantly improves generation quality. Specifically, augmenting the decoder with inferred global predicate values leads to performance gains of $16.7\%$ in DS, $11\%$ in GPTScore, and $11\%$ in MMD, alongside a substantial reduction in training loss after convergence. Further incorporating the dynamic latent state inference captures instantaneous dependencies among predicates, enabling finer-grained causal reasoning and resulting in additional improvements. Similarly, the multi-round LLM rule refinement module also yields consistent gains across all metrics.

## C.10 SCALABILITY AND TIME EFFICIENCY

To evaluate the time efficiency and scalability of our proposed method, we conduct experiments on the synthetic dataset Syn@5 with varying sample sizes: 1000, 3000, 5000, 7000, and 9000. Due to the incorporation of the neuro-symbolic reasoning layer and multi-round LLM rule refinement, our model exhibits slightly lower time efficiency compared to other deep neural methods. However, this does not constitute a substantial gap—training time per epoch remains comparable to baseline models across all dataset sizes.

Encouragingly, the additional complexity of our reasoning mechanism yields tangible benefits. Consistent with the findings in Analysis 3 in the main context and the results in Appendix. C.6, our model maintains strong performance in few-shot settings, which can also be found in Fig. 10. When

the training sample size is reduced to only 1000 samples, our approach significantly outperforms all baseline models across all three generative metrics: KL, QQ-RMSE, and MMD. Moreover, performance improves steadily as sample size increases, demonstrating favorable scalability.

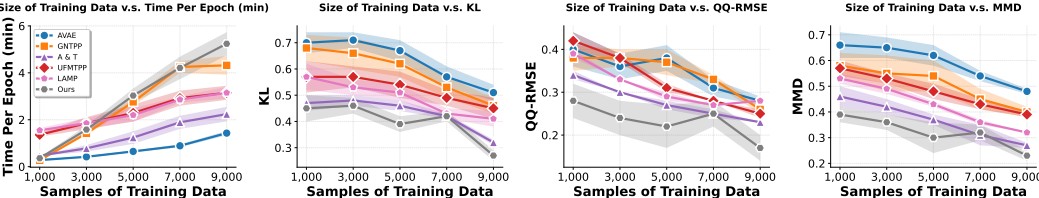

Figure 10: Time efficiency and scalability experiments for Syn@5 dataset. We vary sample size in $\{1000, 3000, 5000, 7000, 9000\}$.

# D REPRODUCIBILITY ANALYSIS

## D.1 HYPER-PARAMETER SELECTION

Our model is easy to implement and reproduce the results. We present the selected hyper-parameters on synthetic, semi-syntheti and real-world datasets in Tab. 14. The hyper-parameter selection metric is a trade-off between training converged loss, generation performance, and time efficiency.

| Hyper-Parameters | Value Used | | | | | |
|---|---|---|---|---|---|---|
| | **Syn@5** | **Syn@10** | **LogiCity** | **MIMIC-IV** | **EPIC-100** | **IKEA ASM** |
| Max Epochs | 32 | 32 | 50 | 50 | 64 | 64 |
| Max Reason Steps ($H$) | 3 | 3 | 7 | 5 | 5 | 5 |
| Batch Size | 32 | 32 | 32 | 32 | 64 | 32 |
| # Heads | 4 | 4 | 4 | 4 | 4 | 4 |
| Attn Embed Size | 32 | 32 | 64 | 32 | 32 | 32 |
| Predicate Embed Size | 32 | 32 | 64 | 32 | 32 | 32 |
| LLM | – | – | – | GPT-4o | GPT-4o | GPT-4o |
| Warm-up Learning Rate | False | False | True | True | True | True |
| Learning Rate | 1e-3 | 1e-3 | 1e-2 | 1e-2 | 1e-2 | 1e-3 |
| Optimizer | Adam | Adam | Adam | Adam | Adam | Adam |

Table 14: Descriptions and values of hyper-parameters used for models trained on the synthetic, semi-synthetic and real-world datasets.

## D.2 COMPUTING INFRASTRUCTURE

All synthetic data experiments and real-world data experiments, including the comparison experiments with baselines, are performed on Ubuntu 20.04.3 LTS system with Intel(R) Xeon(R) Gold 6248R CPU @ 3.00GHz, 227 Gigabyte memory.

## D.3 COMPUTATIONAL AND MEMORY COMPLEXITY

**Computational Complexity** The forward pass complexity is dominated by the encoder and the reasoning layer:

- History encoder: a Transformer over $N$ events: $\mathcal{O}(N^2 \cdot d_e)$
- Neuro-symbolic reasoning layer: the cost per sequence is $\mathcal{O}(|\mathcal{F}| \cdot H \cdot c)$, where $|\mathcal{F}|$ is the number of rules, $H$ is the number of reasoning hops, and $c$ is the average rule body size. This scales linearly with the key parameters.
- Event decoder: a feed-forward network applied at each of the $\hat{N}$ generated events: $\mathcal{O}(\hat{N})$.

**Memory Complexity** The memory footprint is primarily determined by storing. The overall memory usage is linear in the sequence length and model dimensions:

- Sequence embeddings: $\mathcal{O}(N \cdot d_e)$ for the encoder outputs

- Reasoning states: $\mathcal{O}(H \cdot K)$, where K is the number of predicates

## E  USE OF LLMS

In this paper, LLMs were used solely for writing polishing; all substantive writing and content remain human-authored.

## F  LIMITATION AND BROADER IMPACTS

**Limitaion**  Our model relies on LLM-generated symbolic rules as priors, making its performance contingent on the quality, coverage, and prompt design of these rules. Biased or incomplete rules may adversely affect the generation process. Moreover, zero-shot evaluation depends on LLM-based judges, which raises concerns about circularity and the potential reinforcement of pre-existing biases. Finally, the quality of the generated data requires validation on high-quality rare-disease datasets—which are often difficult to acquire and typically require collaboration with medical institutions.

**Broader Impact**  Despite its limitations, our framework presents several positive societal impacts. By incorporating human-readable rules into neural generative models, it can produce interpretable synthetic sequences suitable for privacy-sensitive data sharing, rare-disease data augmentation, and debugging of high-stakes predictive models. However, potential risks emerge if generated data are deployed without rigorous validation: erroneous or biased rules may produce implausible or harmful samples, LLM-based evaluation could overestimate real-world reliability, and synthetic sequences might be misused for unverified clinical applications. To mitigate these concerns, future efforts should emphasize expert-involved rule curation, hybrid human–LLM evaluation protocols, full transparency of prompts and rule sets, and safeguards such as memorization checks and differential privacy in synthetic data release.

