# OpenReview forum: "Neuro-Symbolic VAEs for Temporal Point Processes: Logic-Guided Controllable Generation"
_ICLR.cc/2026/Conference — Submitted to ICLR 2026_

### Official Review · Reviewer_B5fr · 2025-10-29

**Soundness:** 3
**Presentation:** 2
**Contribution:** 2
**Rating:** 4
**Confidence:** 1

**Summary:**

This paper proposes NS-VAE-TPP, a neural-symbolic V framework that integrates a differentiable reasoning layer into a VAE for temporal point processes (TPPs). The model combines generative sequence modeling with symbolic domain knowledge, using forward-chaining rules encoded as predicate embeddings and soft logical operators. It aims to generate irregularly-sampled time series logically consistent with expert knowledge. Experiments on clinical datasets show improved interpretability and prediction performance over baselines.

Note: As my expertise is not aligned with this paper, I may not give useful review. If I did not understand it correctly, please AC ignores my review.

**Strengths:**

1. The paper extends differentiable neuro-symbolic reasoning to irregularly-sampled sequence data.

2. The forward-chaining operator is well-defined and mathematically clear, allowing logic-based reasoning to be fully differentiable within the VAE framework.

3.  By grounding domain knowledge rules from medical knowledge bases, the model can provide interpretable reasoning chains (e.g., “renal dysfunction ← cardiovascular instability ∧ electrolyte imbalance”), which is valuable for healthcare applications.

4. Experimental results suggest that the reasoning layer contributes to more consistent and accurate temporal modeling.

**Weaknesses:**

1. I am confused about the technical contribution.

It seems that the differentiable reasoning layer has been proposed in existing works. This study is mainly combine the previous neuro-symbolic reasoning with the temporal sequence mdoeling rather than introducing a fundamentally new reasoning method.

2. The reasoning layer introduces extra computation proportional to the number of rules and hops (H). It may be helpful to analyze how complex this framework is.

**Questions:**

1. How are the symbolic rules obtained, like from knowledge graphs or text corpora?

---

> ### Author Response · Authors · 2025-12-02
> **Response to Reviewer B5fr**
>
> We thank **Reviewer B5fr** for the detailed and valuable comments! These comments would help us improve the quality of the paper.
>
> ### **-- W1: Clarification on novelty**
>
> Our contribution is not a new reasoning operator, but a **new integration** of symbolic reasoning as a **generative prior** for TPPs, enabling **controllable and reliable sequence generation under data scarcity**.
>
> The key novelty is our **reasoning-before-generation** paradigm. Prior work typically applies symbolic reasoning **post-hoc** or only as **output constraints**, which does not enforce consistency in the latent generative process. In contrast, we embed rules directly into the latent space: the differentiable reasoning layer imputes missing predicates and enforces logical consistency before decoding, shaping the entire generative trajectory.
>
> Thus, our main contribution is an **end-to-end neuro-symbolic generative model that supports constraint-aware generation from partial observations**—an underexplored setting in TPPs—and achieves **good model/parameter and data efficiency** (please refer to our response to **Reviewer zxGR, W1**) and **zero-shot/controllability generation (Tab. 9, Appendix C.7 of the revised paper)** by leveraging symbolic knowledge as a structural prior.
>
> ### **-- W2: Analysis of the framework complexity**
>
> Thank you for the suggestion. The complexity of the reasoning layer is straightforward: with $∣F∣$ rules, max hops $H$, and $c$ predicates per rule body, forward chaining costs $O(N \cdot H \cdot c)$, i.e., linear in the key parameters. In practice, $∣F∣$ is small (5–20 rules) and H is low (3–7), keeping the overhead modest.
>
> Empirically, the scalability results (**Fig. 10, Appendix C.10 of the revised paper**) show only a moderate per-iteration increase over neural baselines, and the additional cost is often offset by the data efficiency gained from incorporating symbolic structure (please refer to the few-shot results in **Fig. 4 of the revised paper**, and our response to **Reviewer zxGR, W1**).
>
> The overhead reflects multi-hop, constraint-aware reasoning in the latent space—necessary for generating logically consistent sequences under partial observability. Overall, the linear cost is a deliberate trade-off that directly enables more reliable, controllable, and data-efficient generation.
>
> ### **-- Q1: How are the symbolic rules obtained?**
>
> Our method does **not** rely on external knowledge graphs or text corpora. Symbolic rules are obtained through a **self-contained LLM-driven extraction and refinement pipeline**, summarized as follows:
>
> - **Rule initialization from LLM prompts**: We provide the LLM with a few representative event sequences and a structured prompt (**Appendix B, Fig. 6**). The LLM outputs candidate Horn-clause rules (e.g., *“renal dysfunction ← cardiovascular instability ∧ electrolyte imbalance”*). No external corpora or knowledge graphs are used.
>
> - **Predicate embeddings**: We extract the LLM’s internal embeddings for each predicate to initialize their semantic representations. (For details please refer to our response to **Reviewer opve, W3**)
>
> - **Trainable rule embeddings**: Rule representations $Θ_f$ are trainable, allowing the model to adjust rule strengths using data rather than relying on raw LLM outputs.
>
> - **Iterative refinement**: As shown in **Appendix B, Fig. 7**, sequences that are poorly explained trigger another LLM query to propose improved rules. This bootstrapping expands coverage and improves rule quality.

---

### Official Review · Reviewer_opve · 2025-10-29

**Soundness:** 2
**Presentation:** 2
**Contribution:** 2
**Rating:** 4
**Confidence:** 4

**Summary:**

The paper proposes the Neuro-Symbolic Variational Autoencoder with Temporal Point Processes (NS-VAE-TPP), a framework for logic-aware sequence generation in continuous time. Specifically, the proposed framework combines a temporal point process backbone for modeling event times and types with a novel reasoning layer in the latent space. The authors also conduct experiments to evaluate the performance of the proposed method.

**Strengths:**

[+] The paper proposes a reasoning-before-generation architecture to embed symbolic rules as generative priors in the latent space.

[+] The proposed method enforces logical consistency and imputes missing structures.

[+] The paper conducts experiments on synthetic, semi-synthetic, and real-world datasets.

**Weaknesses:**

[-] It is unclear about the potential and definitions of symbolic constraints. The papers says that synthetic sequences must also satisfy symbolic constraints-such as eligibility, exclusions, ordering, and timing-that encode dependencies beyond surface correlations. The discussions of all the potential symbolic constraints are no provided. Additionally, the math definitions of these symbolic constraints are not explicitly listed, which makes it hard to understand these symbolic constraints. Further, could the authors discuss all the potential dependencies and how to distinguish between dependences and surface correlations?

[-] The paper injects a few example sequences and contextual instructions to inject domain knowledge. However, the selection and generation of them are not given. Additionally, how many typical example sequences and contextual instructions are needed in experiments are needed? How to ensure the quality of these example sequences and contextual instructions? What is the complexity of generating these example sequences and contextual instructions?

[-] In the proposed method, the authors employ an LLM as the knowledge initializer. However, large language models are known to suffer from hallucination issues, which may lead to the generation of inaccurate or misleading knowledge. Additionally, the confidence or uncertainty of the LLM’s generated outputs can vary significantly across different instances, potentially introducing inconsistency into the initialization process. However, the authors fail to discuss these. Further, it is unclear about how to extract predicates from the LLM’s internal representations.

[-] In the proposed method, evidence from the body predicates is aggregated into a rule score using a differentiable approximation of logical AND. However, during the aggregation process, the different reality degrees of evidence are not considered. Additionally, it is unclear why a differentiable approximation is needed, and there is no discussions on the approximation error in the proposed method.

[-] The authors adopt a fixed number of Hops as an approximation to multi-hop approximation. It is unclear about why this is an effective approximation and whether there are other solutions. What are the potential advantages and disadvantages of this approximation? Additionally, the paper approximates the posterior by using a factorized Bernoulli distribution. Could the authors discuss the potential advantages and disadvantages of this approximation.

[-] The paper does not provide a complexity analysis of the proposed method. Given that the proposed framework involves multiple sequential steps and components, it is important to analyze both the computational and memory complexity to assess its scalability and practicality.


[-] The proposed method appears to perform poorly during the initial stage, where events are sparse or missing (first 0–25%). This limitation suggests that the method may rely heavily on the long-time events. The discussions on potential strategies to enhance robustness when early event information is not given.

**Questions:**

[1] In Eqn. (2), the forward chaining process iteratively adds new facts until no additional facts can be derived. Could the authors clarify the potential maximal number of newly generated facts in this process? What happens if new facts continue to emerge indefinitely? Moreover, for the generated facts, is there a filtering mechanism to remove low-quality or noisy facts? If so, what criteria are used for filtering? Finally, could the authors elaborate on the concept of multi-hop reasoning and how it is integrated within the forward chaining process?

[2] For the dynamics of the TPP, are there any other potential ways to describe these dynamics? The paper talks about the conditional intensity function. However, there is no discussions on why only this conditional intensity function is used to describe these dynamics.

[3] For the generative perspective, could the authors clarify the survival function, and the role of the survival function in Eqn. (3)? Additionally, could the authors clarify the closed-form inverse? Further, could authors clarify in which cases the integrated perspective does not have a closed-form inverse? Do the conducted experiments in the paper always have the closed-form inverses? If not, could the authors provide some ablation studies for this?

[4] Could the authors list the potential hyperparameters in different steps of the proposed method? For these hyperparameters, could the authors briefly talk about the sensitivity of the proposed method to these hyperparameters?

---

> ### Author Response · Authors · 2025-12-02
> **Response to Reviewer opve Part-1**
>
> We thank **Reviewer opve** for the detailed analysis and insightful comments, which benefit us to further improve our paper! To address your concerns, we have prepared a detailed point-by-point response below.
>
> ### **-- W1: Definition of symbolic constraints and distinction between logical dependencies and surface correlations**
> The symbolic constraints in our model are formalized as first-order logic rules (specifically Horn clauses, i.e. $P_0 \leftarrow P_1 ∧ P_2 ∧ ⋯ ∧ P_c$ as defined in **Eq. 1 of Section 2.1**) which serve as a general mathematical formalism for representing such constraints. With regard to the specific types of constraints mentioned by the reviewer, they can be instantiated as follows:
>
> - **Eligibility**: an event $e_i$ can occur only if a prerequisite event $e_j$ has occurred (e.g., Treatment(x) ← Indication(x)).
> - **Exclusion**: events $e_i$ and $e_j$ cannot co-occur (e.g., ¬Treatment(x) ← Contraindication(x)).
> - **Ordering**: events must follow a strict temporal order $t_i < t_j$.
> - **Timing**: specify inter-event intervals (e.g., “lab test → treatment within 24h”).
>
> **Distinguishing dependency vs. surface correlation**. Surface correlations arise purely from co-occurrence in data. Symbolic dependencies in our model are explicit logical relations encoded by rules and applied in the **reasoning layer**, not inferred statistically. For instance, a rule such as *“medication administration ← presence of indication ∧ absence of contraindication”* ensures that decisions are guided by clinically meaningful logic rather than incidental correlations.
>
> Empirically, the **leave-one-out rule ablations** (refer to our response to **Reviewer zxGR, W2**) show that removing a rule markedly degrades performance, confirming that the model’s behavior depends on these logical dependencies rather than incidental correlations.
>
> ### **-- W2: Selection, design, and complexity of example sequences and LLM prompts**
>
> We address two aspects:
>
> - **LLM prompt design for rule extraction**: We use a small set of representative sequences from the training data, chosen to cover diverse patterns and common domain scenarios. In practice, **3–5 sequences** suffice to elicit meaningful rules; increasing to 9 yields only marginal improvements (e.g., MIMIC-IV experiments shown in the table below). Additionally, fixed, manually designed templates guide the LLM to output predicates and Horn clauses. They include high-level prompts and are fixed per domain to ensure consistency. The quality of these instructions is validated empirically through the generated rules' alignment with domain knowledge, and the iterative refinement process (**Fig. 7**) further mitigates any initial imperfections.
>
> | # Example Sequences | 1 | 3 | 5 (Current Setting) | 7 | 9  |
> |-|-|-|-|-|-|
> | DS | 0.38 | 0.32 | 0.31 | 0.30 | 0.31 |
> | GPT-Score | 0.67 | 0.73 | 0.73 | 0.73 | 0.74 |
> | MMD | 0.49 | 0.45 | 0.43 | 0.43 | 0.41 |
>
> - **Conditional generation process**. In the generation phase, after the model is well-trained, our model can synthesize new sequences conditioned on partial observations (e.g., an initial sequence $x_0$). As outlined in **Algorithm 1, Appendix A**, the encoder maps $x_0$ to a latent state, and the reasoning layer performs forward chaining to infer high-level predicates. The decoder then generates subsequent events auto-regressively, ensuring logical consistency with $x_0$. This conditional generation does **not require additional computational overhead** beyond standard inference, as it leverages the pre-trained model **without retraining**. The quality of the generated sequences is inherently robust due to the model's training on diverse data, and thus no special assurance for $x_0$'s quality is needed.

---

> ### Author Response · Authors · 2025-12-02
> **Response to Reviewer opve Part-2**
>
> ### **-- W3: Robust rule extraction and predicate embedding from LLMs**
> **$[-]$ Mitigating LLM hallucination and uncertainty**
>
> Our framework is designed to be robust to this in several key ways:
>
> - **Iterative refinement, not one-time reliance**: Initial LLM-generated rules are not used as ground truth. A **multi-round refinement (Fig. 7, Appendix B)** updates the rule set based on sequences poorly explained by the initial output, improving rule coverage and accuracy.
>
> - **Trainable rule embeddings**: Rule embeddings $Θ_F$ are trainable parameters. During end-to-end training, the model learns to adjust how these rules are applied and combined, effectively fine-tuning the initial LLM output and correcting for spurious or inaccurate rules. The model learns to "trust" useful rules and ignore poor ones. Our response to **Reviewer zxGR, W1** empiracally shows our model can learn and refine rules from data even when LLM-provided rules are incomplete, while maintaining generation quality.
>
> - **Empirical validation**: Performance metrics (DS, GPTScore, MMD; **Tab. 1**) show high-quality generation. Cross-LLM checks (**Appendix C.8**) demonstrate consistent results across different LLMs (e.g., GPT-4o vs OPT-1.5B...). Leave-one-out ablations (refer to our response to **Reviewer zxGR, W2**) confirm that each extracted rule contributes meaningfully. Even with incomplete rules, the model maintains performance (**Reviewer zxGR, W1 and W2**). This indicates that the method does not rely on a single LLM’s idiosyncrasies.
>
> **$[-]$ Predicate extraction from LLM internal representations**
>
> The process for obtaining the predicate embeddings $θ_P$ is as follows:
>
> - Provide the LLMs with the **predicate name** (e.g., "Renal Dysfunction") in the context of a structured prompt.
>
> - Extract the **hidden state vector** of the final layer corresponding to the predicate token(s), capturing semantic meaning.
>
> - Optionally, reduce dimensionality via PCA.
>
> ### **-- W4: Differentiable rule aggregation with evidence-weighted inference**
> **$[-]$ Necessity of a differentiable approximation**
>
> The core reason is to **enable learning and refining the symbolic rules from data**. While the LLM provides an initial rule prior, the rule embeddings $Θ_F$ are trainable parameters. The differentiable forward-chaining operator allows gradients from the generative loss to propagate backward, enabling the model to **fine-tune** the application of the initial LLM-generated rules and **adaptively learn new, data-driven rule patterns** that the LLM might have missed.
>
> If the rules were static, a non-differentiable logic engine would suffice. However, the key innovation is that our neuro-symbolic layer supports both **logical inference** (via forward chaining) and **parameter learning** (via backpropagation), allowing the symbolic knowledge to be dynamically adjusted and improved end-to-end. The predicate embeddings remain fixed anchors, while the rule embeddings evolve during training, which is why differentiability is crucial.
>
> **$[-]$ Handling varying "reality degrees" of evidence**
>
> Aggregation (**Eq. 7**) uses a softmin function weighted by cosine similarity between predicate embeddings and rule slots. Predicates with higher similarity and stronger activation contribute more to the rule score ($u_f$), effectively differentiating strong vs. weak evidence. This provides a principled, differentiable mechanism to integrate uncertain or partial beliefs into rule evaluation.
>
> ### **-- W5: Rationale and trade-offs for fixed hops and factorized Bernoulli posterior**
> **$[-]$ Fixed number of reasoning hops $(H)$**
>
> We use a fixed $H$, which balances **reasoning depth and training stability**. Full forward chaining to a fixed point can require many, instance-dependent steps, causing **gradient instability**. Fixed $H$ ensures uniform computation, prevents vanishing/exploding gradients, and simplifies implementation.
>
> On MIMIC-IV, rule activations saturate beyond $H=5$; further increases yield only marginal improvements, confirming effective approximation.
>
> | H | 1 | 3 | 5 (Current Setting) | 7 | 9 |
> |-|-|-|-|-|-|
> | DS | 0.40 | 0.34 | 0.31 | 0.30 | 0.29 |
> | GPTScore | 0.64 | 0.68 | 0.73 | 0.73 | 0.74 |
> | MMD | 0.51 | 0.46 | 0.43 | 0.45 | 0.42 |
>
> **$[-]$ Factorized Bernoulli posterior**
>
> The rationale is that the latent vector $v$ represents the activation of $K$ high-level Boolean predicates ($v \in [0,1]^K$). A Bernoulli distribution is the natural and simplest probabilistic model for such binary random variables.
>
> This factorization brings several advantages. It supports Gumbel-Sigmoid reparameterization, enabling low-variance, differentiable sampling ($v \sim q(v|x)$) for end-to-end training. The closed-form KL in ELBO (**Eq. 18**) ensures tractable, stable optimization without high-variance Monte Carlo estimates.

---

> ### Author Response · Authors · 2025-12-02
> **Response to Reviewer opve Part-3**
>
> ### **-- W6: Complexity analysis**
>
> We thank the reviewer for this valuable suggestion. We have added a detailed complexity analysis in **Appendix D.3 of the revised paper**. Below is a summary for the key computational and memory costs of our model.
>
> **$[-]$ Computational complexity**
>
> The forward pass complexity is dominated by the encoder and the reasoning layer:
> - History encoder: a Transformer over $N$ events: $O(N^2 \cdot d_e)$
> - Neuro-symbolic reasoning layer: the cost per sequence is $O(|F| \cdot H \cdot c)$, where $|F|$ is the number of rules, H is the number of reasoning hops, and $c$ is the average rule body size. This scales linearly with the key parameters.
> - Event decoder: a feed-forward network applied at each of the $\hat{N}$ generated events: $O(\hat{N})$.
>
> **$[-]$ Memory complexity**
>
> The memory footprint is primarily determined by storing. The overall memory usage is linear in the sequence length and model dimensions:
> - Sequence embeddings: $O(N \cdot d_e)$ for the encoder outputs
> - Reasoning states: $O(H \cdot K)$, where $K$ is the number of predicates
>
> **$[-]$ Scalability and practicality**
>
> The linear scaling of the reasoning layer with respect to $H$ and $∣F∣$ ensures practicality, as these are typically small in practice (e.g., $H \leq 7$, $∣F∣ \leq 20$; see **Tab. 14, Appendix D.1 of the revised paper**). Our empirical results on real-world datasets (e.g., **Fig. 10 of the revised paper**, training time per epoch on MIMIC-IV is comparable to models without complex reasoning components) confirm the framework's efficiency.
>
> ### **-- W7: Limitation of the autoregressive paradigm**
>
> We thank the reviewer for this insightful observation. The reviewer is correct that our model, like all auto-regressive models, faces challenges when generating the very first events with no historical context. This is a **fundamental limitation of the autoregressive paradigm**, not a specific shortcoming of our architecture.
>
> Our primary goal in conducting the initial-stage missingness experiment (**Fig. 5 of the revised paper**) was to **quantify this expected performance lower bound and empirically demonstrate how severe this universal issue is for our model under extreme data scarcity**. The results show that while performance degrades as expected, our method still achieves competitive performance since the MMD (0.32) is relatively low and outperforms most baselines.
>
> To enhance robustness in such scenarios, potential strategies can be considered:
> - Incorporating a global prior: using a learned distribution over plausible sequence beginnings to initialize the generation process.
> - Hierarchical modeling: a "plan-then-detail" approach could be adopted. We can first generate a high-level event skeleton, then refine it autoregressively into a full event sequence.
> - Explicit initial event generation: an auxiliary task can be introduced to generate the first N most likely events from global context. This offers improved guidance at the beginning of decoding.

---

> ### Author Response · Authors · 2025-12-02
> **Response to Reviewer opve Part-4**
>
> ### **-- Q1: Forward chaining, multi-hop reasoning, and fact filtering**
>
> **$[-]$ Maximal new facts and indefinite generation**
>
> The maximal number of new facts is theoretically bounded by the **finite size** of the global predicate set $K$.
>
> **$[-]$ Filtering low-quality facts**
>
> Instead of a post-hoc filter, our model employs a built-in, differentiable filtering mechanism during reasoning. The belief strength of each new fact $v_P$ is computed via a weighted aggregation of rule scores (**Eqs. 7-8**), where the weights are determined by the semantic alignment ($cos(z_i^*, θ_i)$) and current activation of body predicates. This design inherently suppresses low-quality facts by assigning them a low confidence score directly in the continuous latent state $v^{h}$.
>
> **$[-]$ Multi-hop reasoning integration**
>
> Multi-hop reasoning refers to the iterative derivation of higher-level knowledge:
> - Hop 1: derives direct consequences from the initially observed predicates ($v^{0}$).
> - Hop 2: uses the facts from Hop 1 as new premises to infer further conclusions.
> - Repeats for $H$ hops, allowing the model to infer complex dependencies that are not immediately obvious from the raw observations.
>
> ### **-- Q2: Justification for using conditional intensity to model TPP dynamics**
>
> The conditional intensity function is the standard and most fundamental formalism for temporal point processes, as established in the foundational literature [1, 2]. We adopt it because it aligns naturally with the goals of our generative framework and bring following benefits:
>
> - The intensity gives a complete and interpretable description of event likelihoods conditioned on history, providing a principled base for generative modeling.
> - It readily captures rich history-dependent dynamics such as triggering and inhibition, offering far greater expressiveness than fixed-rate or memoryless alternatives.
> - It yields a simple log-likelihood form that integrates cleanly into the ELBO of the VAE framework, consistent with most modern neural TPP models.
>
> While alternative formulations exist (e.g., modeling the survival function or joint density of inter-event times), they are less practical for deep generative settings. The survival function complicates integration with event types, while direct joint density modeling struggles with complex history dependencies.
>
> [1] Rasmussen JG. Lecture notes: Temporal point processes and the conditional intensity function. arXiv preprint arXiv:1806.00221. 2018 Jun 1.
>
> [2] Daley DJ, Vere-Jones D. An introduction to the theory of point processes: volume II: general theory and structure. New York, NY: Springer New York; 2008.
>
> ### **-- Q3: Role and computation of the survival function in generative modeling**
> **$[-]$ Role of the survival function in Eq. 3**
>
> In the generative perspective, $p(t | \mathcal{H}) = \lambda(t | \mathcal{H})S(t | \mathcal{H})$, where the survival function $S(t | \mathcal{H}) = \exp(- \int_0^t \lambda(s | \mathcal{H}) ds)$ ensures that the density is normalized. **Eq. 3** corresponds to this standard decomposition.
>
> **$[-]$ Closed-form inverse**
>
> The closed-form inverse refers to sampling time points via inverse-transform sampling: $t = F^{-1}(u), F(t) = 1 - S(t)$.
> Thus, a closed-form inverse exists when the integral $\int_0^t \lambda(s)ds$ can be inverted analytically.
>
> **$[-]$ When the inverse does not exist**
>
> For general neural intensities, $\lambda(t)$ does not yield a closed-form inverse of $F(t)$. This is the case for models with arbitrary MLP-based intensity functions and is a known limitation of the integrated perspective. In such cases, thinning algorithms are typically required [3].
>
> **$[-]$ Do our experiments require a closed-form inverse?**
>
> No. All experiments in our paper use a neural decoder that does **not rely on closed-form inversion**. We use the standard thinning-based sampling procedure [4] (also as done in AVAE, GNTPP, and Neural TPPs), so closed-form inverses are not needed anywhere in our implementation. Since our model never uses closed-form inverses in training or evaluation, an ablation comparing closed-form vs. numerical sampling is not applicable.
>
> [3] Yosihiko Ogata. On lewis’ simulation method for point processes. IEEE transactions on information theory, 27(1):23–31, 1981.
>
> [4] Shchur O, Gao N, Biloš M, Günnemann S. Fast and flexible temporal point processes with triangular maps. Advances in neural information processing systems. 2020;33:73-84.

---

> ### Author Response · Authors · 2025-12-02
> **Response to Reviewer opve Part-5**
>
> ### **-- Q4: Key hyperparameters and sensitivity analysis in our proposed framework**
> The hyperparameters used in our experiments are provided in **Tab. 14, Appendix D.1 of the revised paper**. We report several representative experiments on the MIMIC-IV dataset from three aspects to analyze the sensitivity of our proposed model to the hyperparameters. Overall, experiment results demonstrate the robust performance across a reasonable range of these parameters. Specifically,
>
> - **Reasoning modules**. Refer to our response to **W5**, as the number of reasoning steps $H$ increases, the performance typically saturates ($H=3-5$), as logic in our target domains requires only limited hops for derivation. Moreover, the experiments on the MIMIC-IV dataset with varying rule counts (below table) show that performance improves substantially when increasing the LLM-extracted rules to $>10$, while further expansion yields only minor gains. This saturation indicates that once core domain semantics are covered, our framework effectively leverages and adaptively refines the provided rules, demonstrating its robust adaptability.
>
> | # Extracted Rules | 1 | 5 | 10 (Current Setting) | 15 | 20 |
> |-|-|-|-|-|-|
> | DS | 0.53 | 0.38 | 0.31 | 0.29 | 0.28 |
> | GPTScore | 0.54 | 0.65 | 0.73 | 0.74 | 0.74 |
> | MMD | 0.72 | 0.55 | 0.43 | 0.43 | 0.41 |
>
> - **Embeddings**. We ablated the predicate embedding size and found stable performance across dimensions, with size 32 achieving optimal results. Larger embeddings increased model complexity without substantial obvious improvements, confirming our choice as an efficient trade-off between representational capacity and training cost.
>
> | Predicate Embed Size | 16 | 32 (Current Setting) | 64  | 128 | 256 |
> |-|-|-|-|-|-|
> | DS | 0.34 | 0.31 | 0.30 | 0.33 | 0.31 |
> | GPTScore | 0.68 | 0.73 | 0.73 | 0.74 | 0.73 |
> | MMD | 0.47 | 0.43 | 0.41 | 0.40 | 0.40 |
>
> - **Architecture choices**. Our ablation study on the transformer encoder shows optimal performance with 4 attention heads (DS: 0.31, GPTScore: 0.73, MMD: 0.43). While other configurations yield slightly lower results, they remain satisfactory, demonstrating the architecture's robustness to this hyperparameter.
>
> | # Heads | 1 | 2 | 4 (Current Setting) | 8 | 16 |
> |-|-|-|-|-|-|
> | DS | 0.39 | 0.36 | 0.31 | 0.28 | 0.27 |
> | GPTScore | 0.66 | 0.65 | 0.73 | 0.74 | 0.71 |
> | MMD | 0.52 | 0.48 | 0.43 | 0.40 | 0.42 |

---

### Official Review · Reviewer_zxGR · 2025-11-03

**Soundness:** 3
**Presentation:** 3
**Contribution:** 3
**Rating:** 4
**Confidence:** 2

**Summary:**

This paper proposes the Neuro-Symbolic Variational Autoencoder with Temporal Point Processes (NS-VAE-TPP), a generative model for continuous-time event sequences designed for safety-critical domains.
The proposed model combines a TPP backbone with a novel neuro-symbolic reasoning layer in the latent space. An encoder maps event sequences to high-level predicate variables. A "Symbolic Prior Bank" (SPB), initialized by querying a Large Language Model (LLM), stores predicate embeddings and a set of symbolic rules. Before generation, a differentiable forward-chaining reasoning module refines the latent predicate state, enforcing logical consistency and imputing missing information. A decoder then generates event times and types conditioned on both the temporal history and this reasoning-augmented latent state. The model is trained end-to-end as a VAE by maximizing the ELBO.

**Strengths:**

- The core architectural idea of "reasoning-before-generation"  is interesting. Placing a symbolic reasoning layer directly within the latent space of the VAE , rather than using rules as a post-hoc filter, is a interesting approach to ensuring that generated sequences are internally coherent.

- The model's demonstrated strength in few-shot and zero-shot scenarios is a practical advantage. This supports the hypothesis that the symbolic priors provide a valuable inductive bias, making the model less reliant on large, complete datasets.

**Weaknesses:**

- The central claim of SOTA performance on synthetic and semi-synthetic data is confusing. The paper states its "advantage stems from our approach's explicit utilization of the complete set of ground-truth logic rules". The baselines (AVAE, GNTPP, etc.) are not given this ground-truth information.
- The evaluation methodology for rule-conditioned generation is also confusing. For real-world datasets, symbolic rules are extracted by querying LLMs. Then, to evaluate generation under these rules, the paper uses LLM judges to produce an "R-Score" (Rule Adherence) and "C-Score" (Plausibility). Using an LLM to judge adherence to rules generated by an LLM is not rigorous, even if this is a limitation.

**Questions:**

See weaknesses

---

> ### Author Response · Authors · 2025-12-02
> **Response to Reviewer zxGR Part-1**
>
> We are grateful for **Reviewer zxGR**’s careful review! We hope our following response addresses your concerns.
>
> ### **-- W1: Additional experiments demonstrating data efficiency, parameter efficiency, and robustness of the proposed framework**
>
> The synthetic/semi-synthetic experiments serve to **isolate the performance upper bound of our framework when accurate rules are available**, and to **demonstrate that the proposed neuro-symbolic architecture can effectively leverage such rules for logically consistent generation**. Our main SOTA comparison is on real-world datasets (MIMIC-IV, EPIC-100, IKEA), where no model has oracle rules, and our method still outperforms all baselines.
>
> To address the reviewer’s concern, we added new results on Syn@5 (**see Sec. 4.2, Analysis 2 of the revised paper for details**):
>
> - **Data efficiency**. With only **1,000 (20%)** training samples, MMD rises mildly (0.30 → 0.38) and remains better than the next-best baseline (0.46), matching that baseline’s performance using 5,000 (100%) samples. This shows that injected rules act as an effective structural prior.
>
> | # Samples |  | 1000 |  |  | 3000 |  |  | 5000 (Current Setting) |  |
> |-|-|-|-|-|-|-|-|-|-|
> | Model | KL | QQ-RMSE | MMD | KL | QQ-RMSE | MMD| KL | QQ-RMSE | MMD |
> | AVAE | 0.70 | 0.40 | 0.66 | 0.71 | 0.36 | 0.62 | 0.67 | 0.38 | 0.62 |
> | GNTPP | 0.68 | 0.38 | 0.58 | 0.66 | 0.38 | 0.55 | 0.62 | 0.37 | 0.54 |
> | A & T | 0.47 | 0.34 | 0.46 | 0.48 | 0.30 | 0.42 | 0.46 | 0.27 | 0.37 |
> | UFM-TPP | 0.57 | 0.42 | 0.57 | 0.57 | 0.38 | 0.51 | 0.54 | 0.31 | 0.48 |
> | LAMP | 0.57 | 0.39 | 0.52 | 0.53 | 0.33 | 0.49 | 0.51 | 0.29 | 0.43 |
> | Ours* | 0.45 | 0.28 | 0.38 | 0.46 | 0.24 | 0.36 | 0.38 | 0.22 | 0.30 |
>
> - **Parameter efficiency**. As shown in **Fig. 3**, baselines require substantially more parameters to approach our results. Below is the parameter count for each baseline used to obtain the results reported in **Tab. 1**. Under the same data budget, models such as AVAE and GNTPP use larger architectures yet still underperform, indicating reduced parameter demand due to structured priors.
>
> | Model | KL | QQ-RMSE | MMD | # Params |
> |-|-|-|-|-|
> | AVAE | 0.67 | 0.38 | 0.62 | 61.29K |
> | GNTPP | 0.62 | 0.37 | 0.54 | 105.24K |
> | A & T | 0.46 | 0.27 | 0.37 | 65.86K |
> | UFM-TPP | 0.54 | 0.31 | 0.48 | 107.28K |
> | LAMP | 0.51 | 0.29 | 0.43 | 130.27K |
> | Ours* | 0.39 | 0.22 | 0.30 | 33.62K |
>
> - **Robustness to incomplete/no rules**. On Syn@5 with **no ground-truth rules**, the model learning rules from scratch achieves 0.54/0.30/0.36 (KL/QQ-RMSE/MMD), still outperforming AVAE (0.67/0.38/0.62) and GNTPP (0.62/0.37/0.54). The framework thus has the ability to adaptively recover useful logic constraints directly from data.
>
> | # Oracle Rules | 0 | 1 | 2 | 3 | 4 | 5 (Current Setting) |
> |-|-|-|-|-|-|-|
> | KL | 0.54 | 0.50 | 0.44 | 0.45 | 0.43 | 0.39 |
> | QQ-RMSE | 0.30 | 0.29 | 0.30 | 0.27 | 0.23 | 0.22 |
> | MMD | 0.36 | 0.38 | 0.35 | 0.35 | 0.32 | 0.30 |
>
> - **Ablation on reasoning hops**. Without oracle rules, performance stabilizes around $H=4$; larger $H$ yields only marginal changes. This indicates that moderate multi-hop reasoning improves generation even when no rules are provided.
>
> | H | 1 | 2 | 3 | 4 | 5 |
> |-|-|-|-|-|-|
> | KL | 0.57 | 0.58 | 0.54 | 0.42 | 0.43 |
> | QQ-RMSE | 0.36 | 0.30 | 0.30 | 0.25 | 0.24 |
> | MMD | 0.43 | 0.38 | 0.36 | 0.33 | 0.32 |
>
> In summary, these additional experiments show that logic rules substantially **reduce the need for large datasets and highly parameterized models**. Even without explicit rules, our model can still **adaptively learn** useful patterns and outperform neural TPP baselines, which require more data or larger models to approach similar performance. **We have added a new “Analysis 2” in the revised paper (in Sec. 4.2) to include these results and analysis**.

---

> ### Author Response · Authors · 2025-12-02
> **Response to Reviewer zxGR Part-2**
>
> ### **-- W2: Mitigating circularity in LLM-based rule extraction and evaluation**
> We agree that using LLMs for both rule extraction and evaluation raises concerns about circularity. This is unavoidable in real-world datasets where **no human-verified rules or deterministic checkers exist**. To mitigate this limitation, we rely on **LLM-free distributional metrics (MMD, KL, QQ-RMSE)** for all core evaluations. And rule extraction and evaluation use **different LLMs**, and extracted rules are **not used verbatim**—they are refined through **trainable rule embeddings**, reducing sensitivity to any specific LLM output.
>
> To further address this concern, we have included the following robustness experiments and have highlighted them more clearly in **Appendix C.8 of the revised paper**:
>
> - **Robustness across multiple LLM judges**.
> As added in **Appendix C.8** and following tables, we evaluate with **multiple LLM judges** (GPT-4o and smaller open-source models). Results show **consistent scores with low variance**, indicating that evaluation is **not driven by a single model’s bias**.
>
> (Note: using multiple LLMs to evaluate the performance on MIMIC-IV, extending the results we reported in **Tab. 1**.)
> | LLM as Judges | GPT-4o | GPT-4o+Opt-1.5B | GPT-4o+Opt-1.5B+Opt-125M | GPT-4o+Opt-1.5B+Opt-125M+Gemma-2 | GPT-4o+Opt-1.5B+Opt-125M+Gemma-2+TinyLlama |
> |-|-|-|-|-|-|
> | GPTScore | 0.72 +/- 0.00 | 0.73 +/- 0.01 | 0.74 +/- 0.01 | 0.72 +/- 0.02 | 0.72 +/- 0.02 |
>
> (Note: using multiple LLMs to evaluate the zero-shot generation ability with pre-defined rules focusing on drug effect, extending the results we reported in **Tab. 9, Appendix C.7**.)
> | LLM as Judges | GPT-4o | GPT-4o+Opt-1.5B | GPT-4o+Opt-1.5B+Opt-125M | GPT-4o+Opt-1.5B+Opt-125M+Gemma-2 | GPT-4o+Opt-1.5B+Opt-125M+Gemma-2+TinyLlama |
> |-|-|-|-|-|-|
> | R-Score | 0.67 +/- 0.00 | 0.69 +/- 0.02 | 0.69 +/- 0.02 | 0.68 +/- 0.03 | 0.68 +/- 0.02 |
> | C-Score | 0.69 +/- 0.00 | 0.71 +/- 0.02 | 0.70 +/- 0.02 | 0.71 +/- 0.02 | 0.71 +/- 0.02 |
>
> - **Rule importance validated by ablations**. The **systematic ablation confirms the collective importance of the extracted rules**. Removing any single rule (in a **leave-one-out way**) causes significant performance degradation, while using just one rule yields the worst results (DS increased by (↑) 71%, GPTScore decreased by (↓) 26%, MMD increased by (↑)  67% vs. full rule set). This demonstrates that all rules are collectively essential for maintaining generation quality in real-world settings.
>
> (Note: removed some rules extracted from LLMs to validate the importance. We take experiments on MIMIC-IV as an example.)
> | # Rules Used | 1 | 3 | 5 | 7 | 10 (Current Setting) |
> |-|-|-|-|-|-|
> | DS | 0.53 | 0.46 | 0.38 | 0.35 | 0.31 |
> | GPT-Score | 0.54 | 0.62 | 0.65 | 0.70 | 0.73 |
> | MMD | 0.72 | 0.60 | 0.55 | 0.49 | 0.43 |
>
> - **Rule importance varies substantially across sequences**. For instance (we only consider delete the rules presented in **Tab. 6, Appendix C.5 of the revised paper**), in Seq-1, removing Rule-4 causes the most severe performance drop, demonstrating its critical role in capturing this specific sequence's pattern.
>
> | Sequence ID | Delete Rule-1 | Delete Rule-2 | Delete Rule-3 | Delete Rule-4 | Delete Rule-5 | No Delete |
> |-|-|-|-|-|-|-|
> | Seq-1 | 0.54 | 0.57 | 0.52 | 0.60 | 0.53 | 0.48 |
> | Seq-2 | 0.41 | 0.41 | 0.38 | 0.46 | 0.55 | 0.36 |
> | Seq-3 | 0.43 | 0.48 | 0.45 | 0.46 | 0.45 | 0.42 |
>
> These results strengthen the methodological soundness of the rule-conditioned evaluation despite the unavoidable lack of human-curated symbolic supervision. The extracted rules also show empirical importance, indicating the effectiveness of our reasoning-layer and LLM extractor.

---

### Author Response · Authors · 2025-12-03
**General Response**

Dear Esteemed Reviewers, Area Chairs, Senior Area Chairs and Program Chairs,

We extend our deepest gratitude to all the reviewers for their dedication and valuable time spent in assessing our paper. Your constructive critiques and insightful suggestions have been instrumental in elevating the quality of our research.

We are particularly thankful for the positive and enlightening feedback from **Reviewer zxGR** on our paper's novelty *"the core architectural idea of reasoning-before-generation is interesting."*. Additionally, we are happy to hear our paper *"extends differentiable neuro-symbolic reasoning to irregularly-sampled sequence data"*, and *"the forward-chaining operator is well-defined and mathematically clear"* from **Reviewers B5fr**. The acknowledgment of our proposed method *"enforces logical consistency and imputes missing structures"* by **Reviewer opve** is also greatly appreciated.

We want to address that our **core contribution** lies in the **reasoning-before-generation** paradigm. We introduce an **end-to-end neuro-symbolic generative model that supports constraint/logic-aware generation from partial observations**—an underexplored setting in TPPs—and achieves **good model/parameter and data efficiency**. **Experimentally**, our method achieves strong state-of-the-art performance across synthetic, semi-synthetic, and real datasets: on Syn@10 it achieves the lowest 0.62/0.53/0.55; and on MIMIC-IV it improves DS by 18%, GPTScore by 14%, and MMD by 23% over the best baseline. The model further demonstrates **strong few-shot performance** (e.g., with 20% MIMIC data outperforming all baselines) and **zero-shot rule-conditioned generation** with high R-/C-scores (0.63–0.70).

In our response, we have further enhanced the manuscript with the following major improvements:

- **Highlight data and parameter efficiency (Reviewer zxGR)**: We added Syn@5 experiments showing strong data efficiency (e.g., our MMD 0.38 with 20% data still surpasses baselines at 100%) and parameter efficiency (33.6K params vs. 61–130K for baselines while achieving lower KL/QQ-RMSE/MMD).

- **Robustness evaluations for LLM judges (Reviewer zxGR, opve)**: We added robustness evaluations showing the model performs well even with incomplete/no LLM rules (e.g., 0.54/0.30/0.36 KL/QQ-RMSE/MMD on Syn@5 still outperforming baselines), cross-LLM judging robustness (GPTScore variance <0.02 across GPT-4o, OPT, Gemma, TinyLlama), and rule ablation studies demonstrating rule importance.

- **Justification for the selection of Hyperparameters (Reviewer opve)**: We further added new analyses on reasoning hops, rule-count sensitivity, predicate embedding size, and transformer-head robustness, showing the trade-off when selecting the hyperparameters.

- **Complexity analysis (Reviewer opve, B5fr)**: We expanded complexity analysis, including computational complexity, memory complexity, and practicality, confirming the efficiency of our framework.

- **Technical clarification (Reviewer opve, B5fr)**: We clarified how are the symbolic rules obtained, why differentiable rule aggregation, rationale and trade-offs for fixed hops and factorized Bernoulli posterior, and justification for using conditional intensity to model TPP dynamics, making the technical details clearer.

We believe our comprehensive enhancements **further strengthen** an already strong contribution. The manuscript now has clarified the data and parameter efficiency of our method (**§4.2 Analysis 2**), validated robustness (**Table 2, Table 11, and Table 12**), expanded clearer complexity analysis (**Appendix D3**), and provided complete technical justification (**§3.3, §3.4**).

Finally, your feedback and guidance during the review process have been invaluable. Once again, we express our heartfelt thanks for your substantial contributions to this review process.

With warm regards,

The Authors

---

### Meta-Review · Area_Chair_5wBH · 2025-12-12

**Summary:**

The reviewers showed interest in this work but overall thought this was below the bar for publication. There wasn't much overlap in terms of the concerns raised by reviewers and two reviews had very low confidence. I hence reviewed the paper myself, and found that most concerns were relevant, and I especially agreed with those of reviewers opve and zxGR.

Based on this, I support the concerns around the choice of the hyper-parameters in the architecture. Beyond the choice of the number of predicates and the number of hops, I am also questioning the choices of AND and OR approximations and whether these can capture the complexity of the rules mentioned in the illustrations. For instance, a few rules mention "not contra-indication" but it is not clear how the negation in the rule can be learned from the described architecture. I am not familiar with the literature on differential logic reasoning and found it difficult to understand what was novel compared to what was common practice as the citations in the methods are scarce.

I found the validation of the method to be limited and would have liked more in-depth analyses of the results, e.g. looking at how well predicates were matching the embeddings, which rules were identified for each dataset, ... I also agree with reviewer opve that comparison with other baselines could have been expanded, especially given the overlap with other architectures such as concept bottleneck models (surprisingly absent from the scholarship while concepts and predicates have a similar semantic meaning). I found that the metrics derived from LLMs were not very convincing and proving utility of the method in a downstream task (e.g. improving early detection of sepsis in MIMIC by adding synthetic data) would have been more convincing.

**Reviewer Concerns:**

The authors have added a few experiments to address the main concerns raised by the reviewers. I believe the concern on model complexity was sufficiently resolved.

While they have varied some of the hyper-parameters of the architecture, it feels like a search over $K$ and $H$ (and maybe other choices) would be necessary for each dataset, which could be prohibitive. The authors also mention vanishing or exploding gradients to justify a fixed $H$, but this issue has multiple solutions that could have been explored (e.g. skip connections, which could make sense as some predicates can be relevant at different hops but not necessarily be important for all steps).

Overall, I found that the response was limited in its scope, only varying hyper-parameters on one synthetic dataset independently of each other. Given the limitation of evaluations in the paper, I believe more ablations would be needed, e.g. by providing curves over hyper-parameters for different datasets and searches to understand the variability of the architecture and rules. More relevant metrics would also be valuable.

While I understand that space is limited and the paper includes multiple datasets and baselines, the full 10 pages are used but there is no limitation section or discussion in the main text, which I believe is an issue.

**Reviewer Scores:**

Based on the concerns described above and the limited additional evidence for the method, I don't believe the paper would have been supported for acceptance.

---

### Decision · Program_Chairs · 2026-01-26

Reject